# DocGenome: A Large Benchmark for Multi-Modal Language Models in Real-World Academic Document Understanding

## Abstract

Scientific documents record research findings and valuable human knowledge, comprising a vast corpus of high-quality data. Leveraging multi-modality data extracted from these documents and assessing large models' abilities to handle scientific document-oriented tasks is therefore meaningful. Despite promising advancements, large models still perform poorly on multi-page scientific document extraction and understanding tasks, and their capacity to process within-document data formats such as charts and equations remains under-explored. To address these issues, we present DocGenome, a structured document benchmark constructed by annotating 500K scientific documents from 153 disciplines in the arXiv open-access community, using our custom auto-labeling pipeline. DocGenome features four key characteristics: *1) Completeness*: It is the first dataset to structure data from all modalities including 13 layout attributes along with their LaTeX source codes. *2) Logicality*: It provides 6 logical relationships between different entities within each scientific document. *3) Diversity*: It covers various document-oriented tasks, including document classification, visual grounding, document layout detection, document transformation, open-ended single-page QA and multi-page QA. *4) Correctness*: It undergoes rigorous quality control checks conducted by a specialized team. We conduct extensive experiments to demonstrate the advantages of DocGenome and objectively evaluate the performance of large models on our benchmark. DocGenome is available at https://anonymous.4open.science/r/DocGenome.

## 1 Introduction

Extracting data from scientific documents and developing large models to understand them is crucial for advancing AI-assisted scientific exploration and discovery (Jumper et al., 2021; Evans et al., 2021; Baek et al., 2021). On one hand, scientific documents provide comprehensive, high-quality, logically rich corpora for training large models (Lv et al., 2023; Chen et al., 2023; 2024; OpenAI, 2023). On the other hand, the ability of large models (Lv et al., 2023; Chen et al., 2023; 2024; OpenAI, 2023) to accurately understand scientific documents is considered as a crucial evaluation criterion.

However, we observed that current Multi-modal Large Language Models (MLLMs) (Li et al., 2020; Zhong et al., 2019; Pfitzmann et al., 2022; Da et al., 2023; Wang et al., 2023b; Chen et al., 2023; 2024; Bai et al., 2023; Alayrac et al., 2022; Li et al., 2023; Tian et al., 2024; Wang et al., 2024c;d; Wu et al., 2023; Zhang et al., 2023; Zhu et al., 2023) still struggle to understand the content of scientific documents as deeply as humans do. This challenge is primarily due to the inherently complicated multi-modal information present in scientific documents, such as multi-modal charts (Xia et al., 2024), intricate equations (Wang et al., 2024a), and sophisticated logical relationships. Currently, MLLMs cannot effectively parse and comprehend such complicated modalities and logical relationships. To alleviate this challenge, we present DocGenome, an open large-scale scientific document benchmark constructed using the designed DocParser.

DocParser is a training-free auto-labeling pipeline, which can generate both attribute information of component units and logical relationships between units by auto-annotating and structuring a large amount of unlabeled arXiv papers , with four stages: 1) data preprocessing, 2) unit segmentation, 3) attribute assignment and relation retrieval, and 4) color rendering as elaborated in Sec. 3.1. Furthermore, we utilize the proposed DocParser to label 500K scientific documents collected from the arXiv open-access community, and the resulting auto-annotated dataset is termed as DocGenome

Figure 1: **Overview of the DocGenome dataset.** Our work introduces DocGenome, a multi-modal dataset of academic documents encompassing 8 primary disciplines, 153 secondary disciplines, 13 categories of component units, and 6 types of entity relationships between units. We showcase an example of the paper (Vaswani et al., 2017) parsing into structured graph forms, termed as the document's genome, by leveraging the attributes and relationships of component units.

(illustrated in Fig. 1), which contains 153 scientific disciplines and 7 document-oriented tasks including: document classification, visual grounding, open-ended single-page and multi-page QA tasks, document layout detection, Equation-to-LATEX transformation, Table-to-LATEX transformation, which is elaborated in Sec. 4.3. Furthermore, we employ the quality grading and human validation methods to ensure the data quality as described in Sec. 3.2 and Sec. 4.2, respectively.

We conduct extensive experiments on the proposed DocGenome benchmark to objectively evaluate many mainstream MLLMs, including QWen-VL Bai et al. (2023), CogAgent Hong et al. (2023), InternVL (Chen et al., 2024; OpenGVLab, 2024), GPT-4V OpenAI (2023), GPT-4o (OpenAI, 2024) and *etc*. The experiments on DocGenome also verify the effectiveness of the proposed dataset, demonstrating its ability to enhance the document understanding of the existing baseline models.

Our main contributions can be summarized as follows:

- For the first time, we construct an open large-scale dataset that includes **500K** structured scientific documents with **13** categories of component units and **6** types of logical relationships between them. This dataset also encompasses various data types within scientific documents, such as Figure, Equation, Table, Algorithm, List, Code, Footnote, and *etc*.

- To construct DocGenome, we design DocParser to automatically generate rich annotation information from the source code of a wealth of arXiv papers.

- DocGenome covers **7** document-oriented tasks, such as document layout detection, document transformation, multi-page QA, *etc*. Besides, we conduct extensive verification and experiments based on these tasks to demonstrate that DocGenome can significantly enhance the document understanding capabilities of the existing baselines.

## 2 RELATED WORKS

**Visual Document Datasets.** To comprehensively show the advantages of the proposed DocGenome dataset, we have reviewed visual document datasets and summarized them in Table 1. In earlier years, visual document datasets (Li et al., 2020; Zhong et al., 2019; Pfitzmann et al., 2022; Da et al., 2023) mainly aim to recognize the region categories of different regions from a given document, such as text region, table region, abstract region, and *etc*. For example, DocBank (Li et al., 2020) constructs 500K high-quality document pages to enable the document layout model to utilize both textual and visual information. Recently, some research works (Mathew et al., 2021; Xia et al., 2023; 2024; Van Landeghem et al., 2023; Li et al., 2024; Liu et al., 2024a) are proposed to build a document dataset with the enhanced diversity from multiple tasks, multiple modalities, and large-scale training data. By comparison, our DocGenome demonstrates more comprehensive features, including the number of disciplines and training samples covered, types of tasks, evaluation metrics, and entity relationships.

Table 1: Comparison with document-related benchmarks. " - " indicates that the corresponding part is not mentioned in the original paper. " * " means that each sample in their training set is cropped from the entire page, resulting in a total of 6.4M samples at the region level rather than the page level.

| Datasets | # Discipline | # Category of Component Units | # Pages in Train-set | # Pages in Test-set | # Task Type | # Used Evaluation Metric | Publication Period | With-Entity Relation |
|---|---|---|---|---|---|---|---|---|
| DocVQA (Mathew et al., 2021) | - | N/A | 11K | 1K | 1 | 2 | 1960-2000 | ✗ |
| DocLayNet (Pfitzmann et al., 2022) | - | 11 | 80K | 8K | 1 | 1 | - | ✗ |
| DocBank (Li et al., 2020) | - | 13 | 0.45M | **50K** | 3 | 1 | 2014-2018 | ✗ |
| PubLayNet (Zhong et al., 2019) | - | 5 | 0.34M | 12K | 1 | 1 | - | ✗ |
| VRDU (Wang et al., 2023c) | - | 10 | 7K | 3K | 3 | 1 | - | ✗ |
| DUDE (Van Landeghem et al., 2023) | - | N/A | 20K | 6K | 3 | 3 | 1860-2022 | ✗ |
| $D^4LA$ (Da et al., 2023) | - | **27** | 8K | 2K | 1 | 3 | - | ✗ |
| Fox Benchmark (Liu et al., 2024a) | - | 5 | N/A (No train-set) | 0.2K | 3 | 5 | - | ✗ |
| ArXivCap (Li et al., 2024) | 32 | N/A | 6.4M* | N/A | 4 | 3 | - | ✗ |
| **DocGenome (ours)** | **153** | 13 | **6.8M** | 9K | **7** | **7** | 2007-2022 | ✓ |

**Automated Document Annotation Tools.** PaperMage (Lo et al., 2023) is an automated annotation tool based on LayoutParser (Shen et al., 2021), which primarily utilizes detection models and OCR tools to annotate research document PDF. S2ORC (Lo et al., 2020) deponds on GROBID (GRO, 2008–2024), which consists of various trainable modules (such as segmentation models, detection models, and text extraction models, etc.) to convert literature PDFs into XML format. By comparison, our DocParser is training-free; it processes LaTeX source code directly without relying on trainable models (such as detection and segmentation models), thus eliminating the need for additional data to support training.

**Visual Document Understanding.** Research in the field of document Artificial Intelligence (AI) has made rapid progress, due to its successful applications in visual document layout analysis (Wang et al., 2023a; Van Landeghem et al., 2023; Da et al., 2023; Appalaraju et al., 2024; Luo et al., 2024; Huang et al., 2022; He et al., 2023b) and image representation learning (Zhou et al., 2024; He et al., 2022; Dosovitskiy et al., 2020; Bengio et al., 2013). Inspired by Transformer (Vaswani et al., 2017), LayoutLMv3 (Huang et al., 2022) utilizes word-patch features to perform pre-training and designs a cross-modal alignment for document AI. UDIO (Tang et al., 2023) tries to unify multiple document-oriented vision tasks using task-specific prompting. Besides, Kosmos-2.5 (Lv et al., 2023) generates the text outputs by a shared decoder-only Transformer. mPLUG-DocOwl (Ye et al., 2023) boosts the OCR-free document understanding ability. Recently, ICL-D3IE (He et al., 2023a) proposes an in-context-based learning framework to integrate LLM into document information extraction tasks and LayoutLLM (Luo et al., 2024) employs the layout instruction mechanism to improve the ability of document analysis.

**Multi-modal Large Language Models (MLLMs).** The development of MLLMs has profound impacts on the Artificial General Intelligence (AGI) landscape. Recently, commercial MLLMs (OpenAI, 2023; Team et al., 2023; Anthropic, 2024; Reid et al., 2024) have experienced extremely rapid progress. GPT-4V (OpenAI, 2023) has significantly advanced the MLLMs. Google's Genimi series (Team et al., 2023; Reid et al., 2024) further enhance the ability of MLLMs to process text, images, and audio. Besides, open-source MLLMs (Wang et al., 2023b; Chen et al., 2023; 2024; Bai et al., 2023; Alayrac et al., 2022; Lu et al., 2024; Li et al., 2023; Lin et al., 2024; Liu et al., 2023; Sun et al., 2023; Tian et al., 2024; Wang et al., 2024c;d; Wu et al., 2023; Zhang et al., 2023; Zhu et al., 2023) have also attracted great attention. Such MLLMs bring accessibility to the rapid development of AI, enabling widespread multi-modal applications and fostering innovation across industries.

## 3 DATA COLLECTION METHODOLOGY FOR DOCGENOME

### 3.1 INTRODUCTION OF AUTO-LABELING PIPELINE

In this section, we present DocParser, a cutting-edge auto-labeling pipeline that streamlines the extraction of labeled source code from unlabeled arXiv data, serving as a key instrument for annotating the DocGenome dataset. As shown in Fig. 2, the annotation process of DocParser is concisely divided into four stages, mitigating the issues of data scarcity and annotation expenses.

**Stage 1: Data Preprocessing.** Our primary focus is to improve the data quality and enhance the compilation success rate of LaTeX source code. Initially, we undertake an expansion of all files referenced by the `\input` and `\include` commands, followed by a series of crucial pre-processing steps. These steps encompass the integration of requisite environment packages, the exclusion of

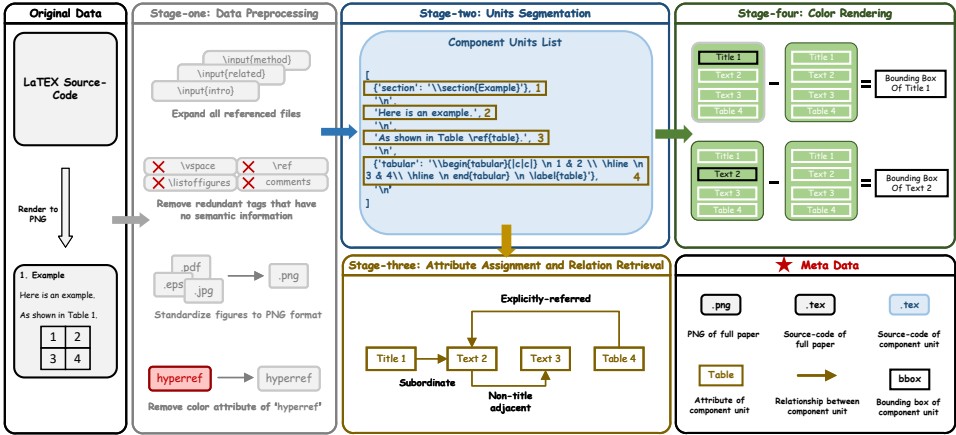

Figure 2: **Schematic of the designed DocParser pipeline for automated document annotation.** The process is divided into four distinct stages: 1) Data Preprocessing, 2) Unit Segmentation, 3) Attribute Assignment and Relation Retrieval, and 4) Color Rendering. DocParser can convert LaTeX source code of a complete document into annotations for component units with source code, attributes, relationships, and bounding box, as well as a rendered PNG of the entire document.

Table 2: The definition of logical relationships between component units.

| Relation Name | Specific Description | Example |
|---|---|---|
| *Identical* | Two units share the same source code. | Cross-column text; Cross-page text. |
| *Title adjacent* | The two titles are adjacent. | (\section{introduction}, \section{method}) |
| *Subordinate* | One unit is a subclass of another unit. | (\section{introduction}, paragraph within Introduction) |
| *Non-title adjacent* | The two text or equation units are adjacent. | (Paragraph 1, Paragraph 2) |
| *Explicitly-referred* | One unit refers to another unit via footnote, reference, etc. | (As shown in \ref{Fig: 5} ..., Figure 5) |
| *Implicitly-referred* | The caption unit refers to the corresponding float environment. | (Table Caption 1, Table 1) |

comment lines, and the removal of extraneous tokens such as \vspace, \ref, and other annotations that do not contribute to the semantic essence of the document. Note that we only remove the \ref that were not compiling correctly (i.e. displaying as "Fig. ??"). Subsequently, we concentrate on standardizing the figure format within the LaTeX source code, converting all graphical elements to the PNG format. Furthermore, we remove the color attribute from the "hyperref", ensuring that the LaTeX source code is ready for targeted color rendering during annotation in stage 4.

**Stage 2: Units Segmentation.** The objective of this phase is to automate the segmentation of content units, thereby streamlining the rendering process for distinct sections. We employ the TexSoup[¶] library to decompose the LaTeX source code into a structured list, delineating each individual component unit. This list is organized according to the reading order, ensuring a logical progression and facilitating the subsequent retrieval of relationships between the component units.

**Stage 3: Attribute Assignment and Relation Retrieval.** We have defined **13** fine-grained layout attributes (more details in Table A.1 of Appendix C) for the component units decomposed in Stage 2, encompassing elements such as Algorithms, Captions, Equations, etc. For each unit, we match an appropriate attribute from the predefined set using keyword queries and regularization techniques to ensure a tailored and precise categorization. In the analysis of component unit relationships, units are categorized into two classes: **1) fixed-form units**, including Text, Title, Abstract, etc., which are characterized by sequential reading and hierarchical relationships readily discernible from the list obtained in Stage 2, and **2) floating-form units**, including Table, Figure, etc., which establish directional references to fixed-form units through commands like \ref and \label. The comprehensive set of **6** entity relationships is detailed in Table 2.

---

[¶]TextSoup package: https://github.com/alvinwan/TexSoup.

**Stage 4: Color Rendering.** The bounding box of a component unit is an additional label we aim to extract. After the segmentation phase in Stage 2, we render the target unit in black and all other units in white, to create two distinct PDFs. By performing a subtraction operation between these documents, we can obtain the detection box containing only the current unit, as illustrated in the top-right corner of Fig. 2. For component units that traverse across hurdles or pages, we standardize the bounding box labels based on their unified source code information. This method effectively mitigates the issue where bounding boxes may be inadvertently divided, ensuring seamless and unified labeling for such units.

We automate the annotation process by sequentially applying DocParser's four stages and leveraging the complete LaTeX source code. This yields not only the document's PDF but also the individual source code, bounding box, specific attributes for each component unit, and the relationships between units. Together, these elements constitute our DocGenome dataset.

## 3.2 DocGenome Benchmark Analyses

Utilizing the DocParser automated annotation tool, we have annotated a corpus comprising 500K academic articles from the arXiv repository. Our analysis explores the diversity of the DocGenome benchmark, focusing on discipline distribution, content distribution, and quality grading.

**Discipline Distribution.** The DocGenome consists of 8 primary disciplines, which collectively encompass 153 secondary disciplines[‖], reflecting a diverse and extensive coverage of academic research areas. The distribution across these disciplines is detailed in Fig. A.2 of Appendix E.

**Year Distribution.** DocGenome archives articles from arXiv, ranging from 2007 to 2022, with a median publication year of 2016. A significant portion, approximately 32.88%, of these articles have been published since 2020. The distribution of these publications over time is depicted in Fig. 3a.

**Content Distribution.** We have examined two key aspects: the distribution of page counts and the labeling of component units. On the dimension of page counts, the dataset's documents have an average page count of 13, with the longest document reaching 50 pages. The distribution of page counts is graphically represented in Fig. A.1 of Appendix C. Moving to the labeling perspective, we have annotated a substantial collection of 500K documents, totaling 74.5M component units and 68.5M relationship labels. In Fig. 1, we present a detailed visualization of the distribution of both the attribute tags of the component units and the relationship labels.

**Quality Grading.** We establish two metrics to grade the data quality of the auto-labeled data that are generated using our DocParser. The first metric, designated as Eq. 1, measures the overlap among auto-annotated bounding boxes within each paper, thereby evaluating the intra-consistency of annotations:

$$IoU_{\text{intra}} = \frac{1}{N(N-1)} \sum_{i=1}^{N} \sum_{j=1, j \neq i}^{N} J(B_i, B_j), \tag{1}$$

where $J(B_i, B_j) = \frac{O(B_i, B_j)}{A(B_i) + A(B_j) - O(B_i, B_j)}$ is the $IoU$ between bounding boxes $B_i$ and $B_j$. $N$ is the total number of annotated bounding boxes in each paper. $O(B_i, B_j)$ represents the overlap area between bounding boxes $B_i$ and $B_j$. $A(\cdot)$ refers to the area of the bounding box.

Eq. 2 shows the second metric that quantifies the overlap between these annotated bounding boxes and the reference bounding boxes (predicted by DocXChain (Yao, 2023)), providing an assessment of the annotations' alignment with established benchmarks, as formulated in Eq. 2:

$$IoU_{\text{align}} = \frac{1}{N} \sum_{i=1}^{N} J(B_i, G_i), \tag{2}$$

where $G_i$ is the $i$-th reference bounding box generated by DocXChain (Yao, 2023), $B_i$ refers to the bounding box that is closest to $G_i$ within our annotated ones.

A lower $IoU_{\text{intra}}$ with a higher $IoU_{\text{align}}$ indicates a higher quality of auto-annotated bounding boxes. Specifically, we split the collected paper into three tiers based on the annotation results. For the *Tier*-1 set, we select the papers with $IoU_{\text{intra}} < 0.05\%$ and $IoU_{\text{align}} > 60\%$, while those with $0.05\% \leq IoU_{\text{intra}} < 1\%$ and $IoU_{\text{align}} > 35\%$ are packed in the *Tier*-2 set, and the remaining papers

---

[‖]According to the arXiv Category Taxonomy: https://arxiv.org/category_taxonomy.

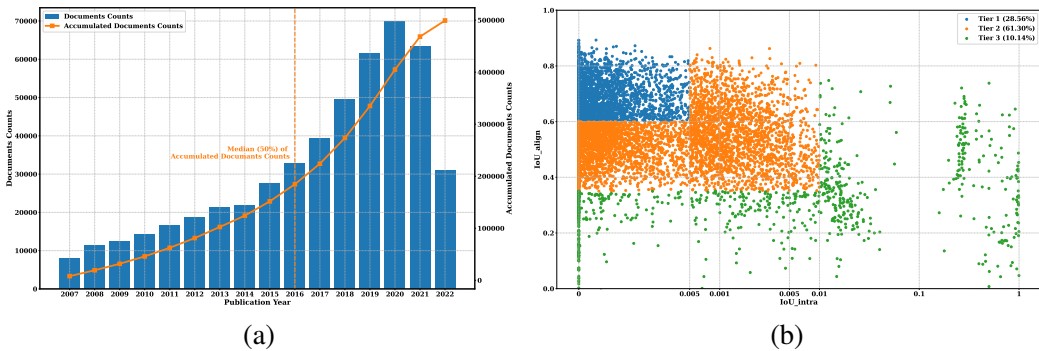

$$(a) \qquad\qquad\qquad (b)$$

Figure 3: **Visualization of data distribution in DocGenome**. (a) Document publication counts over the years. (b) Distribution of three *Tiers* determined by $IoU_{\text{intra}}$ and $IoU_{\text{align}}$.

are categorized as the *Tier*-3 set. The distribution of three-tier data sets is shown in Fig. 3b, indicating that 28.56% of the data was allocated to *Tier*-1, 61.30% to *Tier*-2, and the other 10.14% to *Tier*-3.

## 4 DocGenome-test: A Multi-task, Multi-modal, Comprehensive Evaluation Set for Document Understanding

### 4.1 Principles of Constructing Evaluation Set

We use two principles to split the auto-annotated data into a high-quality evaluation set (**termed as DocGenome-test**) with precise annotation and a large-scale multi-modal training set (**termed as DocGenome-train**). First, the evaluation set should share the same discipline distribution as the collected data. Hence, the test data are uniformly sampled across each discipline. Second, the annotation of test data should be as precise as possible. Therefore, the test data are only sampled from the *Tier*-1 set. Based on these two principles, we finally sampled 1,004 papers (covering 9K pages) as the test set from the overall 500K auto-annotated papers (containing 6.8M pages). As a result, the DocGenome-test covers 1,004 scientific documents with 1K document classification examples, 2K visual grounding examples, 3K QA pairs, 110K layout bounding boxes, 3K Table-L&TEX pairs, and 5K Equation-L&TEX pairs.

### 4.2 QA Pair Generation and Quality Assurance

In the DocGenome-test, we further design multiple Question-Answering (QA) pairs for each paper to comprehensively evaluate the document understanding capabilities of different models. For each paper sampler, two single-page QA pairs and two multi-page QA pairs are generated using GPT-4V (OpenAI, 2023). Specifically, we instruct GPT-4V to randomly select two representative pages, extract useful information from the two pages respectively, and then generate corresponding single-page QA pairs. Additionally, we utilize GPT-4V to search for content-related paragraphs from different pages to construct the cross-page QA pairs, testing the model's ability to understand and integrate information across multiple pages. The QA pairs involve various commonly raised questions whose answers can be precisely inferred from the given paper.

After generating QA pairs for all paper samples in the DocGenome-test, we invited professional faculty members from various fields to conduct the quality assurance checks. Each QA pair is reviewed by three reviewers for cross-verification. The first step involves the initial review by Kimi[††], a well-known paper understanding model, to assess the initial correctness and identify the target location of QA information on the assigned page. Next, based on the provided location of QA information, two professional faculty members are assigned to manually and independently check each QA pair for accuracy, relevance, and clarity. At this stage, the quality evaluation involves the correctness, relevance, and rationality of the designed questions and the accuracy of the provided answer. Finally, the two manually-evaluated results, along with the automatically-evaluated result are cross-verified with the original text to ensure accuracy and consistency. Please refer to Appendix F for more details.

---

[††]Kimi online API: https://kimi.moonshot.cn.

Table 3: Comparison of state-of-the-art multi-modal large language models on the proposed DocGenome-test, including document classification, visual grounding, open-ended single-page, and multi-page QA tasks. Please refer to Sec. 4.4 for the employed evaluation metrics.

| Model | #Params | Classification Acc↑ | Visual Grounding | | Document QA | |
|---|---|---|---|---|---|---|
| | | | Title Edit Distance↓ | Abstract Edit Distance↓ | Single-Page GPT-acc↑ | Multi-Page GPT-acc↑ |
| *Multi-modal Large Language Models* | | | | | | |
| QWen-VL (Bai et al., 2023) | 9.6B | 0.8237 | 0.0775 | 0.8054 | 0.1156 | 0.0627 |
| CogAgent (Hong et al., 2023) | 17.3B | 0.5857 | 0.0166 | 0.5306 | 0.1772 | - |
| DocOwl-1.5 (Hu et al., 2024) | 8.1B | 0.3307 | 0.0509 | 0.6555 | 0.3084 | - |
| Text-Monkey (Liu et al., 2024b) | 10B | 0.7331 | 0.0371 | 0.4551 | 0.1142 | - |
| InternVL 1.5 (Chen et al., 2024) | 26B | 0.7590 | 0.0222 | 0.3601 | 0.4529 | 0.3577 |
| InternVL 2 (OpenGVLab, 2024) | 26B | 0.8855 | 0.0176 | 0.2320 | 0.5019 | 0.4125 |
| GPT-4V (OpenAI, 2023) | N/A | **0.9821** | 0.0096 | **0.0431** | 0.6101 | 0.6501 |
| GPT-4o (OpenAI, 2024) | N/A | 0.9761 | **0.0095** | 0.0654 | **0.7183** | **0.6762** |

## 4.3 EVALUATION TASKS

To comprehensively evaluate the models' understanding capability of scientific documents, we design **7** tasks *w.r.t* each paper document for the DocGenome-test, including document classification, visual grounding, open-ended single-page, and multi-page QA tasks, document layout detection, Equation-to-LaTeX transformation, and Table-to-LaTeX transformation.

Specifically, document classification involves recognizing the field to which a paper belongs. Visual grounding involves identifying the content according to the provided visual components and textual prompts. Document layout detection refers to the localization and recognition of each layout block in given papers. Document transformation encompasses two format conversions, *i.e.*, Table-to-LaTeX and Equation-to-LaTeX transformation. All tasks take the paper images as visual input for inference. The visual examples for each task are illustrated in Fig. A.8 in Appendix I.

## 4.4 EVALUATION METRICS

**Document Classification:** Top-1 Accuracy (%) is used as the metric for document classification tasks, where higher values indicate better performance.

**Visual Grounding:** Edit Distance is used to evaluate the accuracy of visual grounding, with lower values indicating better performance.

**Document Layout Detection:** mAP@0.5:0.95 is evaluated as the metric for document layout detection, where higher values indicate better performance.

**Document Transformation:** We utilize Edit Distance, Jaccard Similarity, Cosine Similarity, and BLEU as metrics to comprehensively evaluate the document transformation task.

**Open-ended QA:** GPT-acc (%) is designed for tasks with open-ended answers, where outputs are evaluated against the ground truth using GPT-4. Please refer to Appendix G for more details.

## 5 EXPERIMENTS

### 5.1 COMPARED BASELINES AND IMPLEMENTATION

**Compared Baselines.** We select various models as baselines for different tasks to provide comprehensive comparisons. Specifically, various multi-modal language models, *e.g.*, QWen-VL (Bai et al., 2023), CogAgent (Hong et al., 2023), DocOwl-1.5 (Hu et al., 2024), Text-Monkey (Liu et al., 2024b), IntenVL 1.5 (Chen et al., 2024), InternVL 2 (OpenGVLab, 2024), GPT-4V (OpenAI, 2023) and GPT-4o (OpenAI, 2024) are tested on document classification, visual grounding, open-ended single-page QA and multi-page QA tasks. For the Document Layout Detection task, we compare DocXChain (Yao, 2023) and YOLOv8 (Jocher et al., 2023). Additionally, we employ Mathpix, a representative commercial software for mathematical formula transformation, as the compared method for the Document Transformation task, including Equation-to-LaTeX and Table-to-LaTeX transformations.

**Implementation Details.** We utilize a combination of document images and instruction prompts as the input. Note that all tasks use a single-page document image as the input, except for the multi-page QA task, which contains at least two consecutive pages of the document. Besides, the multi-page QA

Table 4: Experiments on scaling up the data using the DocGenome-train, with the resulting models evaluated on document layout detection task. We fine-tune YOLOv8 (Jocher et al., 2023) model using the DocGenome-train with different amounts of training data.

| Model | Training Data Amount | mAP@0.5:0.95↑ | Title | Text | Figure | Caption | Equation | Table | Footnote |
|---|---|---|---|---|---|---|---|---|---|
| *Layout detection task on DocGenome-test* | | | | | | | | | |
| DocXChain (Yao, 2023) | N/A | 53.20 | 49.21 | 79.22 | 43.85 | 48.18 | 49.36 | 72.79 | 29.79 |
| YOLOv8 (Jocher et al., 2023) | 7K | 77.47 | 71.79 | 92.48 | 76.29 | 86.56 | 80.65 | 85.81 | 48.43 |
| YOLOv8 (Jocher et al., 2023) | 70K | 89.42 | 83.46 | 95.56 | 86.36 | 94.92 | 90.13 | 92.77 | 82.72 |
| YOLOv8 (Jocher et al., 2023) | 700K | **91.37** | **86.05** | **95.96** | **88.46** | **95.71** | **93.06** | **93.77** | **86.52** |

Table 5: Experiments on scaling up the data using the DocGenome-train, with the resulting models evaluated on equation and table transformation tasks. EqVLM-B and TableVLM-B mean that we train a visual encoder and a text decoder using the DocGenome-train for the equation and table transformation task, respectively.

| Model | Training Data Amount | Edit Distance↓ | Jaccard Similarity↑ | Cosine Similarity↑ | BLEU↑ |
|---|---|---|---|---|---|
| *Equation-to-LaTeX task on DocGenome-test* | | | | | |
| Qwen2VL-7b (Wang et al., 2024b) | N/A | 0.5824 | 0.6979 | 0.5506 | 0.1449 |
| Mathpix[‡] | N/A | 0.4738 | 0.7226 | 0.6045 | 0.4472 |
| EqVLM-B | 10K | 0.3781 | 0.8157 | 0.7840 | 0.5165 |
| EqVLM-B | 100K | 0.2795 | 0.8505 | 0.8317 | 0.5862 |
| **EqVLM-B** | 1M | **0.2111** | **0.8736** | **0.8621** | **0.6352** |
| *Table-to-LaTeX task on DocGenome-test* | | | | | |
| Qwen2VL-7b (Wang et al., 2024b) | N/A | 0.4876 | 0.7598 | 0.6979 | 0.4016 |
| Mathpix[§] | N/A | 0.4436 | 0.7730 | 0.5826 | 0.3528 |
| TableVLM-B | 5K | 0.4821 | 0.8158 | 0.7804 | 0.4596 |
| TableVLM-B | 10K | 0.4738 | 0.8635 | 0.8187 | 0.4973 |
| TableVLM-B | 100K | 0.3091 | 0.8903 | 0.8571 | 0.5340 |
| **TableVLM-B** | 500K | **0.2223** | **0.8997** | **0.8800** | **0.5552** |

task can only be evaluated on the models that support multi-image inputs. For the layout detection task, which uses the single-page document image as input, we use YOLOv8 (Jocher et al., 2023) as the training baseline, trained for 30 epochs with the AdamW optimizer (Loshchilov & Hutter, 2017), with a learning rate of 0.01. For Equation-to-LaTeX and Table-to-LaTeX tasks, we first use the layout annotations to crop out different modalities, *e.g.*, Table, Equation, *etc.*, from the original images. We then employ the same model structure as Pix2Struct-B (0.2B parameters) (Lee et al., 2023) to perform the fine-tuning on DocGenome-train, resulting in EqVLM-B and TableVLM-B. The fine-tuning process lasts for 30 epochs on 64 NVIDIA A100 80G GPUs, with an initial learning rate of 0.00005 and a weight decay of 0.01.

## 5.2 PERFORMANCE ON DOCGENOME-TEST

We evaluate the performance of several state-of-the-art multi-modal large language models on the proposed DocGenome-test, covering document classification, visual grounding, and both single-page and multi-page QA tasks. As shown in Table 3, among the tested models, GPT-4V (OpenAI, 2023) achieves the highest classification accuracy with 98.2% Top-1 Acc, while QWen-VL (Bai et al., 2023) and InternVL 2 (Chen et al., 2024) also show competitive results with 82.4% and 88.6% accuracy, respectively. For the visual grounding task, GPT-4o showcases the best performance in the Title OCR Grounding task with the lowest Edit Distance of 0.0095, while GPT-4V outperforms other models in the Abstract OCR Grounding task with the lowest Edit Distance of 0.0431. In the single-page QA task, GPT-4o attains the highest GPT-acc score of 71.8%, indicating its superior ability to handle document-based QA tasks. For the multi-page QA task, GPT-4o again leads with a GPT-acc score of 67.6%, further demonstrating its robustness in handling multi-page document queries.

## 5.3 EFFECTIVENESS OF DOCGENOME-TRAIN

To validate the effectiveness of the proposed DocGenome-train, we further conduct experiments on scaling up the training data using the DocGenome-train dataset, evaluating the performance improvements of different tasks, *e.g.,* layout detection and document transformation tasks.

Specifically, for the layout detection task, we present the evaluation performance of YOLOv8 (Jocher et al., 2023) under three different training scales in Table 4. It shows that the model's layout detection capacity continually and significantly improves by increasing the training data volume. Regarding the per-attribute performance improvement, the most significant benefit is observed for "Footnote" attribute, which increases from 48.43% to 86.52% mAP after scaling up the training data from 7K to 700K. Compared with DocXChain (Yao, 2023) that only supports the annotation of seven attributes,

Table 6: Document parsing results on Scihub domain, which shows the generalization ability of the proposed DocGenome on other disciplines outside the core focus of arXiv.

| Model | mAP@0.5:0.95↑ | Title | Text | Figure | Caption | Equation | Table | Footnote |
|---|---|---|---|---|---|---|---|---|
| *Layout detection task on Human-annotated data* | | | | | | | | |
| DocXChain (Yao, 2023) | 37.99 | 32.53 | 59.00 | **67.17** | 38.71 | 12.98 | 38.99 | 16.54 |
| YOLOv8-doc (DocGenome) | **50.15** | **42.59** | **64.87** | 56.65 | **64.51** | **47.14** | **47.08** | **28.21** |

| Model | | Edit Distance↓ | Jaccard Similarity↑ | Cosine Similarity↑ | BLEU↑ |
|---|---|---|---|---|---|
| *Equation-to-LaTex task on Sci-Hub data* | | | | | |
| Mathpix[‡] | | **0.4873** | **0.7437** | **0.7295** | **0.1137** |
| EqVLM-B (DocGenome) | | 0.6627 | 0.6303 | 0.5726 | 0.0602 |

Table 7: The evaluation dataset we constructed from the Scihub domain is detailed along the distribution of disciplinary classes, with only a very small proportion of the disciplinary overlapping with the arXiv domain.

| | Medicine | Chemistry | Biology | Humanities | Physics | Engineering | Math | Ecology | Computer Science | Economics | Geography |
|---|---|---|---|---|---|---|---|---|---|---|---|
| Amount | 237 | 159 | 150 | 121 | 84 | 67 | 36 | 35 | 27 | 25 | 25 |
| Proportion | 24.53% | 16.46% | 15.53% | 12.53% | 8.70% | 6.94% | 3.73% | 3.62% | 2.80% | 2.59% | 2.59% |

our trained YOLOv8 consistently outperforms it in seven attributes, validating the effectiveness of the DocGenome-train.

As illustrated in Table 5, for the document transformation task, we conduct similar experiments on Equation-to-LaTeX task and Table-to-LaTeX task, respectively. In these two tasks, we further explore different scaling up settings, with the observation that both tasks benefits the most from scaling up training data from 10K to 100K. Additionally, considering that Edit Distance is more reliable and rigorous to evaluate the similarity, we can observe that the Table-to-LaTeX task has the potential to improve more than the Equation-to-LaTeX task by continuous scaling up. This is because the performance improvement between 100K and 500K training data for TableVLM-B largely exceeds the improvement between 100K and 1M training data for EqVLM-B as shown in Table 5.

## 5.4 Further Discussions

**Generalization on Out-Of-Distribution (OOD) Data.** In this part, we provide a method to extend DocGenome to other disciplines or domains. For disciplines not covered by the arXiv open-access community, such as civil engineering, chemistry, and materials science disciplines, we assume that we can collect their PDF data. Then, we can use the layout detection annotations provided by the proposed DocGenome to train a document parsing model, as demonstrated by the YOLOv8-doc model in Table 6. Our goal is to validate the generalization ability of this model in other disciplines or domains.

Specifically, we select data from the Scihub domain to validate the generalization ability of our models. The detailed discipline information of the constructed Scihub domain is shown in Table 7. Then, we conducted model evaluations on the aforementioned Scihub data, which is annotated by human experts, for the layout detection task and the Equation-to-LaTeX task, respectively. As shown in Table 6, for the layout detection task, the YOLOv8-doc model (Jocher et al., 2023) trained using DocGenome-train presents better generalization ability than DocXChain (Yao, 2023) on human-annotated data. Regarding the Equation-to-LaTeX task, although the performance of EqVLM-B declines on OOD data (Scihub data), it still maintains relatively strong results with an Edit Distance of 0.6627. Considering that Mathpix is closed-source with potential exposure to various data distributions in its commercial usage, it is natural that our trained model performs relatively worse than Mathpix in the OOD data.

**Layout Understanding could Boost QA Performance.** For questions related to figures or tables in DocGenome-test, we directly annotated the detection boxes for figures or tables on the document images to serve as the image input for the QA task. In Table 8, taking InternVL-1.5 as an example, when the images of the papers contain layout information (this information can be considered as a prompt) relevant to the questions, the performance of the QA task can be further enhanced.

**Potential Applications of DocDenome.**

1) The relationships between document elements facilitate the expansion of multimodal data: Utilizing the relation information between entities, we can index associated modal information for specific tasks, enabling data re-annotation and expansion. For instance, in constructing table-related QA tasks, we can not only obtain images of tables but also index the text describing the tables in the document,

Table 8: Comparison of QA performance with and without layout information in document images.

| Model | #Params | Layout Information in Image | Single-Page QA | Multi-Page QA |
|---|---|---|---|---|
| InternVL 1.5 (Chen et al., 2024) | 26B | ✗ | 0.4529 | 0.3577 |
| InternVL 1.5 (Chen et al., 2024) | 26B | ✓ | **0.4922** | **0.4030** |

thereby enriching multimodal information for re-annotation and task expansion. The relationships make our annotation information more flexible and actionable.

2) Enhanced Document Retrieval: Most current RAG methods simply extract text-only information from PDF for retrieval, ignoring the multimodal information from a given document. Our proposed DocGenome contains a large amount of annotated multimodal information, which can be used for performing the multimodal RAG tasks.

3) Automated Research Tools: We can use the Table-LaTeX and Equation-LaTeX pairs from DocGenome to directly train format conversion tools, which facilitates the editing and processing of scientific papers. Moreover, based on the mentioned multimodal RAG capabilities, we can develop an automated tool for summarizing scientific papers, which would make it more convenient and efficient to summarize scientific discoveries.

4) Pioneering Idea Innovator: The automated scientific research tool needs to draw inspiration from interdisciplinary papers across different disciplines, which can be supported by DocGenome's 500K papers across 153 disciplines. In detail, fine-grained annotations support knowledge retrieval, while annotated multimodal data fosters tools for deeper scientific insights, like mathematical proof and table/chart comprehension.

**Representative Ability of LaTeX in Scientific Literature Domain.** The collection using LaTeX is representative of scientific documents, as LaTeX is the preferred tool for academic writing in STEM fields due to its precision and professional formatting. While LaTeX does impose strict formatting rules, these are not constraints but rather mechanisms to ensure accuracy and consistency. For instance, the "??" marker for missing references serves as a clear indicator of errors, prompting authors to address them before finalizing the document. This feature actually enhances the quality of scientific writing by reducing the likelihood of overlooked mistakes. In contrast, other formatting software may not flag such issues, potentially leading to incomplete or inconsistent references. As a result, the strictness of LaTeX contributes to its reputation as a standard for rigorous scientific documentation.

On the other hand, **the arXiv community hosts papers under the CC license**, and all papers are represented in LaTeX format. As a result, the structured scientific literature we obtain based on arXiv (using LaTeX code) also complies with the CC license, which helps to widely promote the dissemination and use of our open-source dataset (DocGenome). Moreover, by leveraging LaTeX code, we can automatically extract annotated structures from 600,000 scientific papers without incurring any additional costs.

## 6 CONCLUSION

In this paper, we introduced DocGenome, a large-scale, structured, multi-task, and multi-modal dataset for scientific documents. We constructed DocGenome using DocParser, our developed auto-labeling pipeline, to extract structured attributes and relationships between units. DocGenome's comprehensive task coverage, logicality, diversity, and correctness make it a valuable resource for training models related to scientific documents and evaluating the capabilities of such large models.

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

## A  OVERVIEW OF APPENDIX

Due to the nine-page limitation of the manuscript, we provide more information on our benchmark and further experiment details from the following aspects:

- Sec. B: Limitations and Dataset Accessibility.
  - Sec. B.1: Limitations.
  - Sec. B.2: Dataset Accessibility.
- Sec. C: Annotation Explanations.
- Sec. D: Annotation in Cross-domain Scenarios.
- Sec. E: More Statistical Distributions of DocGenome.
- Sec. F: Details of Quality Assurance.
- Sec. G: Prompt Design for GPT-acc.
- Sec. H: Annotation Examples in DocGenome.
- Sec. I: Task Examples in DocGenome-test.
- Sec. J: Clarification about prompts utilized when MLLMs are tested on DocGenome.
- Sec. K: Explanation of Model Inference Speed and Resource Consumption

## B  LIMITATIONS AND DATASET ACCESSIBILITY

### B.1  LIMITATIONS

The purpose of our DocGenome is to build a comprehensive scientific document dataset, promoting the development of intelligent document processing and effective evaluation of MLLMs in document understanding tasks. Although our DocGenome provides annotations for 6 categories of entity relationships, exploring the impact of these entity relationship annotations on large models' understanding of scientific documents is highly meaningful. For future works, we will explore the role of the entity relationships in understanding scientific documents.

### B.2  DATASET ACCESSIBILITY

**Dataset Statistics and Analyses:** We have conducted extensive data statistics and analyses, along with thorough quality checks including DocGenome-train and DocGenome-test datasets, which are presented in Sec. 3.2 and Sec. 4.2.

**Long-term Preservation:** To ensure the long-term preservation of the DocGenome dataset, we have uploaded it to Google Drive: https://drive.google.com/drive/folders/1OIhnuQdIjuSSDc_QL2nP4NwugVDgtItD?usp=sharing. This ensures continuous accessibility to the dataset for an extended duration. Furthermore, we will routinely back up the data and monitor its availability to maintain continued accessibility.

**Terms of Use and License:** We have chosen the CC BY 4.0 license for our dataset, as required. This information is included in our paper submission and will also be clearly stated on our dataset website.

**Discussion of Personally Identifiable Information.** All the scientific documents in our DocGenome are sourced from the arXiv open-access community, where papers are released under the CC license. Besides, the arXiv community ensures that papers uploaded by authors adhere to legal and ethical guidelines, including the protection of personal information and the avoidance of offensive material. Thus, we can confirm that our DocGenome does not contain personally identifiable information or offensive content.

## C  ANNOTATION EXPLANATIONS

We provide the annotation details of DocGenome in Table A.1, where the index number in the annotation corresponds to the category index in the attribute list.

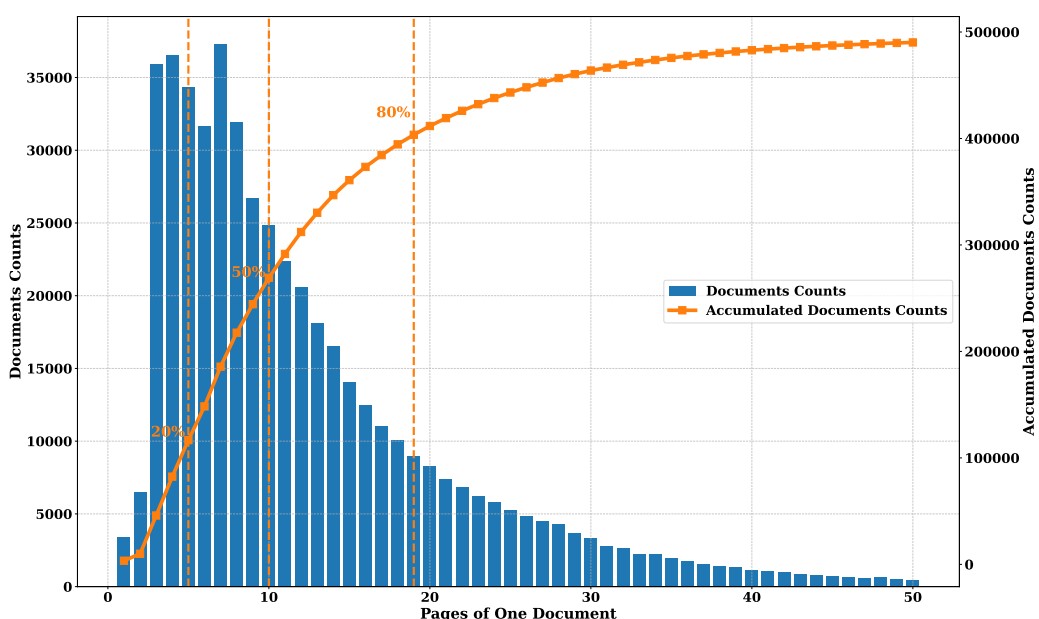

Figure A.1: Page distribution of DocGenome. 20% of documents are five pages or fewer, 50% are ten pages or fewer, and 80% are nineteen pages or fewer.

Table A.1: Category descriptions of the layout annotation performed by our DocParser. Note that we do not use the "others" category and the "reference" category, and their indices are 6 and 11, respectively.

| Index | Category | Notes |
|---|---|---|
| 0 | Algorithm | |
| 1 | Caption | Titles of Images, Tables, and Algorithms |
| 2 | Equation | |
| 3 | Figure | |
| 4 | Footnote | |
| 5 | List | |
| 7 | Table | |
| 8 | Text | |
| 9 | Text-EQ | Text block with inline equations |
| 10 | Title | Section titles |
| 12 | PaperTitle | |
| 13 | Code | |
| 14 | Abstract | |

## D    ANNOTATION IN CROSS-DOMAIN SCENARIOS.

Our DocParser is well-equipped to handle various disciplines on arXiv because we process the LaTeX source code of the papers directly, rather than using a detection model that requires training to annotate paper images. This approach significantly reduces the impact of different writing styles and layout designs across disciplines on automated annotation.

Specifically, we have provided statistics on the quality distribution of papers in different primary disciplines within DocGenome. As described in the main text, we have divided the annotation data into three tiers, and our proposed annotation tool DocParser exhibits similar performance across different disciplinary papers.

---

‡The version of the online API we used for evaluation: https://mathpix.com/equation-to-latex.

§Online API we used for evaluation: https://mathpix.com/table-to-latex.

Table A.2: Distribution of annotation quality (Tier 1, 2, and 3) of different disciplines by DocParser in DocGenome.

| Discipline | Total account | Tier-1 account | Tier-1 proportion | Tier-2 account | Tier-2 proportion | Tier-3 account | Tier-3 proportion |
|---|---|---|---|---|---|---|---|
| cs | 187574 | 65273 | 34.80% | 112950 | 60.22% | 9351 | 4.99% |
| econ | 1679 | 491 | 29.24% | 1037 | 61.76% | 151 | 8.99% |
| eess | 16669 | 5516 | 33.09% | 10432 | 62.58% | 721 | 4.33% |
| math | 20517 | 6579 | 32.07% | 13024 | 63.48% | 914 | 4.45% |
| physics | 250932 | 57328 | 22.85% | 155222 | 61.86% | 38382 | 15.30% |
| q-bio | 2163 | 617 | 28.53% | 1351 | 62.46% | 195 | 9.02% |
| q-fin | 3455 | 1256 | 36.35% | 2022 | 58.52% | 177 | 5.12% |
| stat | 16320 | 5559 | 34.06% | 10036 | 61.50% | 725 | 4.44% |

## E    MORE STATISTICAL DISTRIBUTIONS OF DOCGENOME

In addition to the statistical distribution described in Sec. 3, we provide more statistical distributions in this section. As shown in Fig. A.2, the sample counts of all secondary disciplines are summarized and marked with different colors, from which it can be observed that the inter-discipline and intra-discipline distributions are both diverse, with Physics, Computer Science, and Mathematics papers occupying the major components of DocGenome.

We also present the page distribution of DocGenome in Fig. A.1, which indicates the diversity of paper length in DocGenome. Specifically, 50% papers in DocGenome have nearly or fewer than 10 pages, with 80% papers having fewer than 19 pages.

## F    DETAILS OF QUALITY ASSURANCE FOR QA DATA

**Overall Process of QA Data generation.** 1) GPT-4 was used to generate 7028 questions for 1757 paper samples. 2) Quality checkers first examine the questions, retaining or modifying them to obtain correct questions. 3) Each question was then allocated to two quality checkers for review and correction. 4) The checkers attempted to correct incorrect answers and assigned confidence scores. 5) Only QA pairs with the same answer and the highest confidence scores from both checkers were retained for the final dataset. Finally, 2498 QA pairs were retained to form the QA test set, of which 1672 were modified by the quality checkers.

**The QA Generation Details.** We provide a general prompt template for QA pair generation in Fig. A.3. The discipline-specific guidance is imposed to generate the corresponding ground-truth labels to achieve diversity and relevance.

**The Quality Checking Details.** During independent verification by professional faculty members, each judgment was assigned with a confidence value ranging from 0 to 3. The confidence criterion is designed as follows:

**Confidence 3**: The reviewer is confident that the QA pair is accurate and relevant to the provided paper.

**Confidence 2**: The reviewer thinks the QA pair is mostly accurate and relevant to the provided paper but is unsure whether it is absolutely correct.

**Confidence 1**: The reviewer has no idea about the correctness or relevance of the QA pair to the provided paper.

**Confidence 0**: The reviewer is confident that the QA pair is wrong or irrelevant to the provided paper.

During the cross-verification, the confidence values of the two professional faculty reviewers were compared with the automatically-annotated correctness. The QA pairs with inconsistent results were re-analyzed by the two reviewers and updated to a precise version with consistent confidence.

## G    PROMPT DESIGN FOR GPT-ACC

We adopt GPT-acc as the evaluation metric for the QA tasks. The complete prompts are concluded in Fig. A.4.

## H    EXAMPLES IN DOCUMENT-LEVEL ANNOTATION FROM DOCGENOME

We present one example in DocGenome in Figs. A.5, A.6, and A.7 to visualize the annotations of each page in a whole document (Vaswani et al., 2017). The blocks marked with different colors refer to different attributes of component units and the arrows with different colors denote different relations between units.

## I    EXAMPLES OF TASKS IN DOCGENOME-TEST

We provide visual demonstrations in Fig. A.8 for all 7 tasks in DocGenome-test, including document classification, visual grounding, open-ended single-page and multi-page QA tasks, document layout detection, Equation-to-LaTeX transformation, and Table-to-LaTeX transformation.

## J    CLARIFICATION ABOUT PROMPTS UTILIZED WHEN MLLMS ARE TESTED ON DOCGENOME.

We have concluded the prompts used in Table 3 of the experimental section, where all models were provided with the same prompts for the identical tasks.

- **Classification:** *Which discipline does this article belong to? Select the answer from the given options (%s, %s, %s, %s). Do not print other text.*
- **Visual Grounding (Title):** *Please print the title of this article.*
- **Visual Grounding (Abs):** *Please print the full content of the abstract section of this article directly.*
- **QA:** *question*

## K    EXPLANATION OF MODEL INFERENCE SPEED AND RESOURCE CONSUMPTION.

We have supplemented average model inference speed and resource comsumption for MLLMs when tested on DocGenome in Table A.3. Note that the inference speed indicates the average single-inference speed of MLLMs across all tasks.

Table A.3: Inference speed and memory consumption of MLLMs when tested in DocGenome.

| Model | # Params | Infer Speed (s) | GPU Memory Usage |
|---|---|---|---|
| Qwen-VL | 9.6B | 0.94 | 19685M |
| CogAgent | 17.3B | 4.97 | 37823M |
| Docowl 1.5 | 8.1B | 1.21 | 19005M |
| InternVL 1.5 | 26B | 3.11 | 54019M |
| TextMonkey | 10B | 0.88 | 21417M |

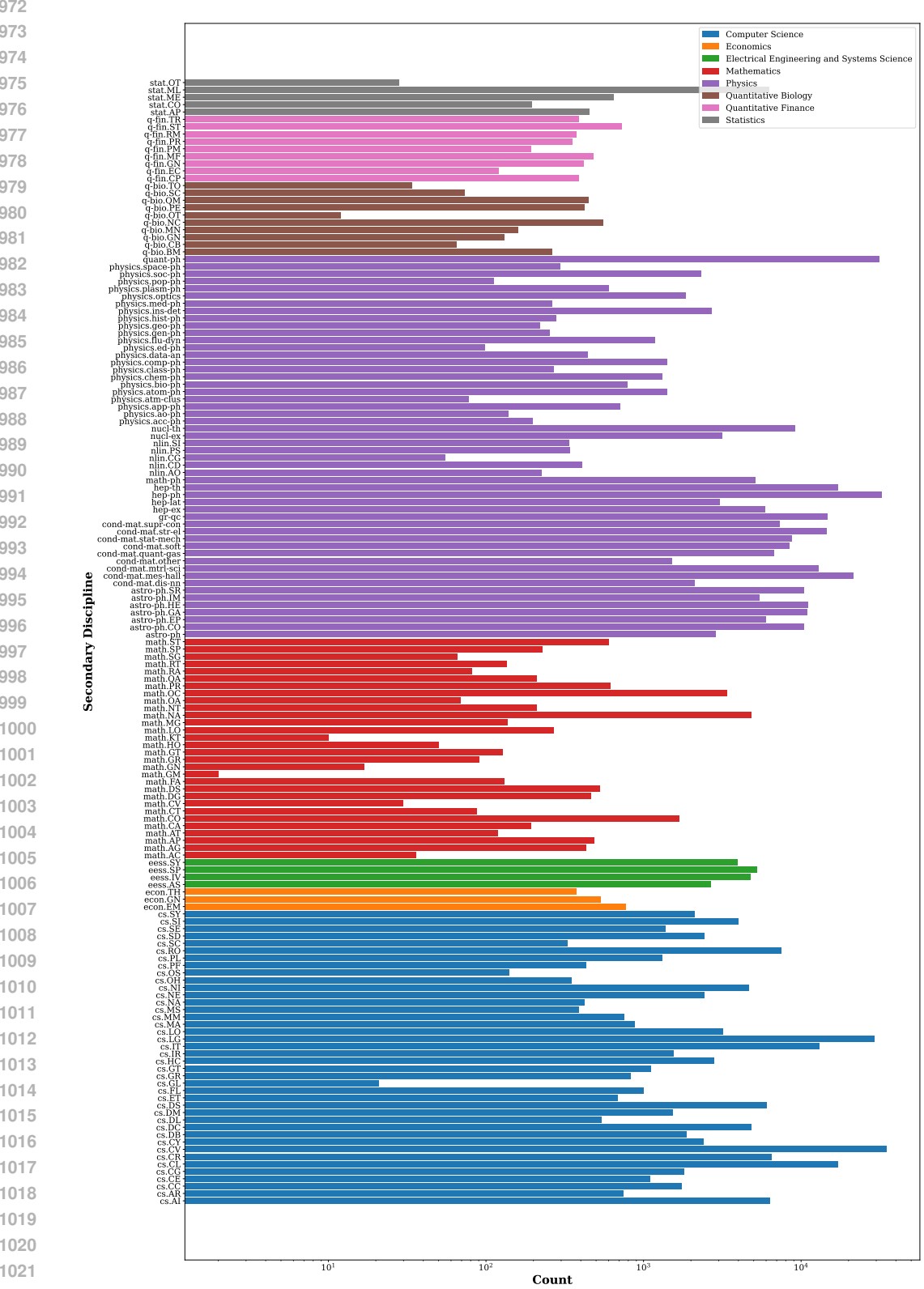

Figure A.2: Distribution of secondary disciplines in our DocGenome. The count on the x-axis represents the number of documents, and documents from the same primary discipline are marked with the same color.

## QA Generation Template

Assume you are an expert in the analysis of arxiv papers. Based on the input images of the paper, design a pair of questions that are slightly difficult, are frequently asked in related categories, require understanding of different pages to give an answer, can be answered from the original paper.
Each answer should not contain any hints, explanations, or notes, etc.
Make sure your answers are accurate. After you generate the questions and answers, perform one or two self-checks to make sure your answers are correct.
Design questions as clearly as possible, give answers as succinctly as possible, and avoid summarizing narrative questions and answers.

The questions should be in the form of a question-answer pair.
Make sure the answer to the question is taken directly from the original text, not from your summary and make sure answers are as short and direct as possible.

Here are some simple examples:
1. Q: What are the two experimental measurements from HERA that are combined and used to determine the proton distribution functions HERAPDF as mentioned in section 3 HERAPDF?
   A: H1 and ZEUS
2. Q: What are the two main types of deep inelastic scattering experiments discussed in the paper?
   A: Inclusive and semi-inclusive
3. Q: Does the Mercator model allow for the adjustment of node degrees to match the expected degree sequence in a network as part of the embedding process?
   A: Yes
4. Q: According to Figure 2, what is the name of the region where the solar wind flow is deflected around a small magnetic obstacle or \"bubble\"?
   A: Narrow barrier region
5. Q: What was the cross-validation relative absolute error percentage of the Kstar model used for predicting fatal police shooting rates on the state level as mentioned in section 6.1?
   A: 28.53%

Please follow this format and give two pairs of answers to the questions.

Figure A.3: Template prompts using GPT-4V (OpenAI, 2023) for document QA pair generation.

## GPT-acc for DocVQA

**Examples:**

```
{
  "query": "<question> What was the incremental increase in revenue from 2020 to 2021? <groundtruth answer> 5 million $ <answer> 20\n",
  "answer": "False"
},
{
  "query": "<question> What percentage of government spending was allocated to infrastructure in 2020? <groundtruth answer> 10% <answer> 14-4=10\n",
  "answer": "True"
},
{
  "query": "<question> What is the total production of Wind Energy in the four months from January to April 2021? <groundtruth answer> 2300 MW <answer> The total production of Wind Energy in the four months from January to April 2021 is 2450 MW.",
  "answer": "False"
},
{
  "query": "<question> What is the value of baseline distance L for the DUNE analysis mentioned in Table I? <groundtruth answer> 1300km <answer> The value of baseline distance L for the DUNE analysis mentioned in Table I is 1300km.",
  "answer": "True"
},
{
  "query": "<question> According to the caption of Figure 5, what is the fixed value of M_N1 used to predict the relic density as a function of m_n? <groundtruth answer> 200 GeV <answer> The fixed value of M_N1 used to predict the relic density as a function of m_n is 200 GeV.",
  "answer": "True"
}
```

**Instruction:**

Given multiple question-answer pairs and the corresponding predictions, evaluate the correctness of predictions. The output should be only "True" or "False"

**Input:**

```
f```
  <question> {question} <groundtruth answer> {answer_gt} <answer> {answer_pred}
```
```

Figure A.4: Detailed prompts in GPT-acc metric for document QA tasks.

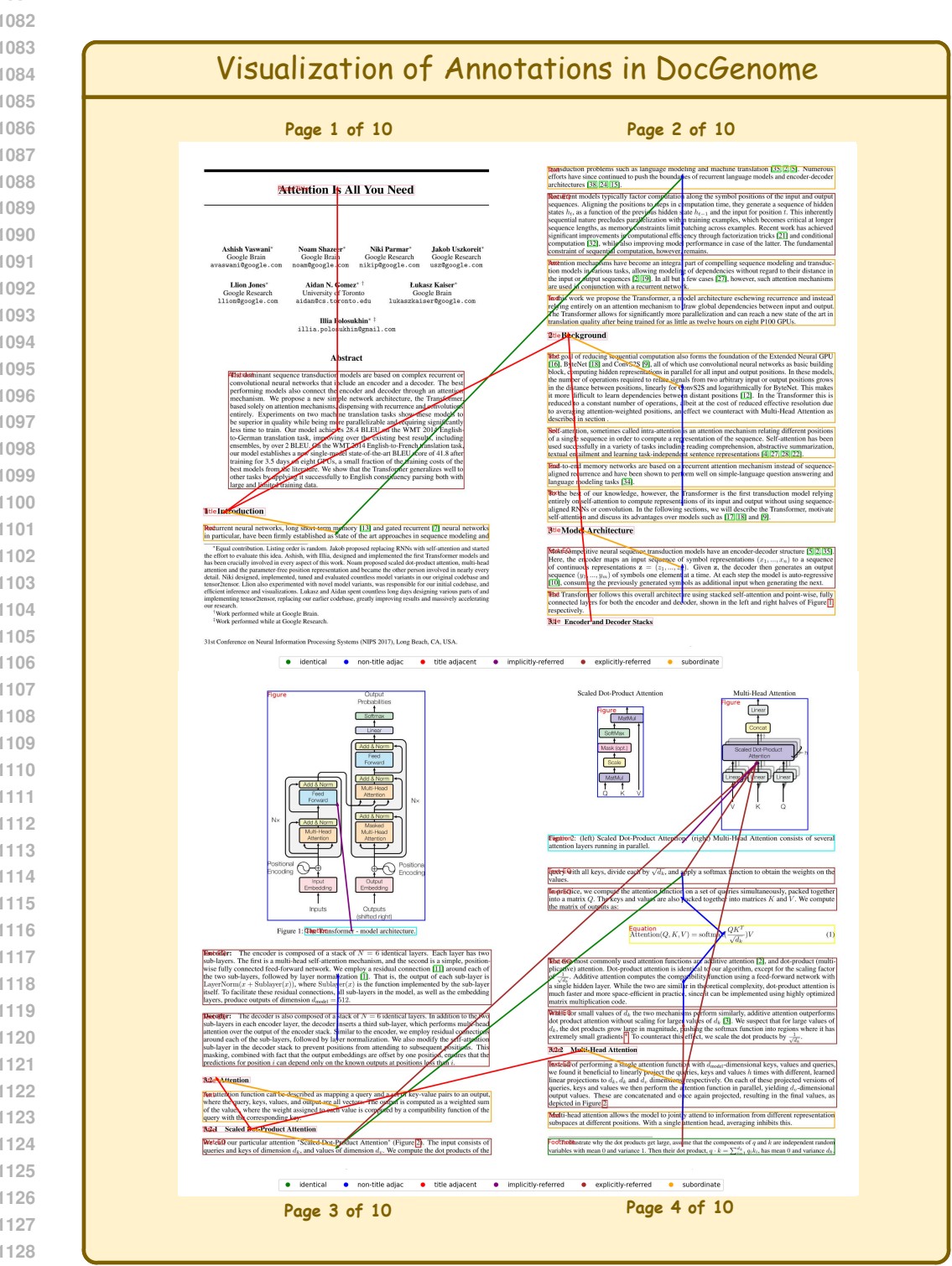

Figure A.5: Annotations of a complete document in DocGenome, taking '*Attention is All Your Need*' (Vaswani et al., 2017) as an example.

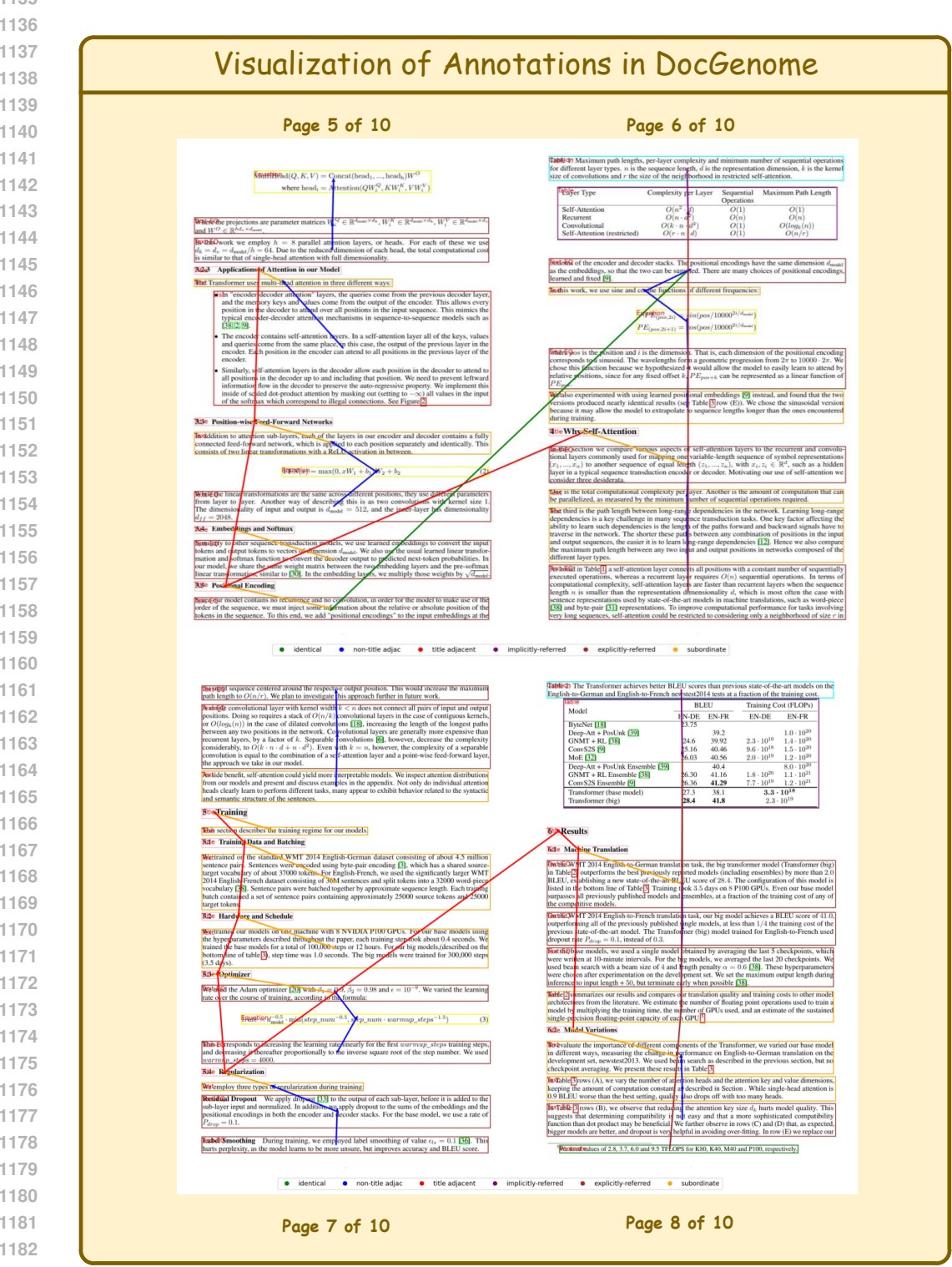

Figure A.6: Annotations of a complete document in DocGenome, taking '*Attention is All Your Need*' (Vaswani et al., 2017) as an example.

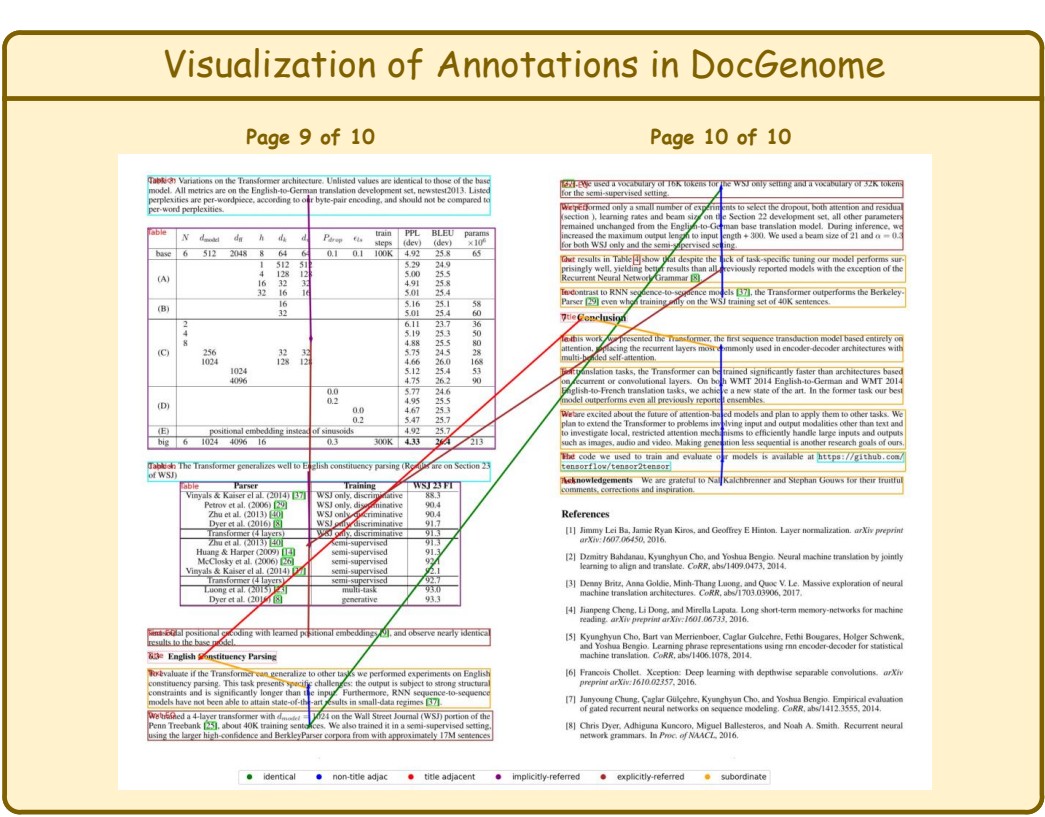

Figure A.7: Annotations of a complete document in DocGenome, taking '*Attention is All Your Need*' (Vaswani et al., 2017) as an example.

## 7 Tasks in DocGenome-test

### 1. Document Classification

Q: Which discipline does this article belong to? Select the answer from the given options (quant-ph, physics.hist-ph, cs.CL,math.PR).

A: quant-ph

### 2. Visual Grounding

Q: Please print the full content of the abstract section of this article.

A: We consider indirect detection of meta-stable dark matter particles decaying into astable neutral particle and a pair of standard model fermions, Due to the softer energy ⋯⋯

### 3. Layout Detection

Title: [232, 448,1416,672]
Abstract: [230,1430, 1469,1877]

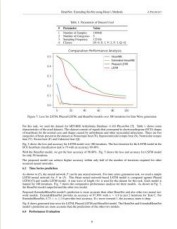

### 4. Single-page QA

Q: What is the best result achieved by the HeunNet model for ECG heartbeat classification?

A: 98.80%

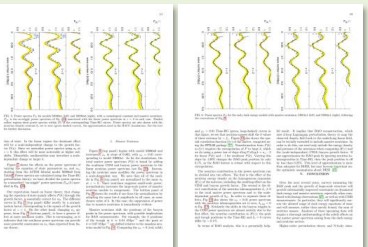

### 5. Multi-page QA

Q: According to Figure 5, what are the shaded yellow regions indicative of in the power spectra P_cb for models M000n1 and M000n2?

A: They show power spectra within 2% of the corresponding Time-RG curves.

### 6. Equation to LaTeX

$$\|A - Q_1 M Q_2^T\|_F^2 = \|A - Q_1 Q_1^T A Q_2 Q_2^T + Q_1 Q_1^T A Q_2 Q_2^T - Q_1 M Q_2^T\|_F^2. \quad (15)$$
$$= \|A - Q_1 Q_1^T A Q_2 Q_2^T\|_F^2 + \|Q_1^T A Q_2 - M\|_F^2.$$

\\begin{equation}\n\\begin{aligned}\n& \\|{\\bf A} - {\\bf Q}_1{\\bf M}{\\bf Q}_2^T\\|_F^2 = \\|{\\bf A} - {\\bf Q}_1{\\bf Q}_1^T{\\bf A}{\\bf Q}_2{\\bf Q}_2^T \\\\ & + {\\bf Q}_1{\\bf Q}_1^T{\\bf A}{\\bf Q}_2{\\bf Q}_2^T \n- {\\bf Q}_1{\\bf M}{\\bf Q}_2^T\\|_F^2\\\\\n ⋯⋯ \\end{equation}

### 7. Table to LaTeX

| Model | L1 | MS-SSIM | Inception | FID |
|---|---|---|---|---|
| Internal benchmark | | | | |
| Non-exemplar | 0.018 | 5.05E-2 | 3.96 | 11.27 |
| Reference | 0.014 | 3.97E-2 | 3.82 | 7.67 |
| Code | 0.015 | 4.15E-2 | 3.94 | 8.49 |
| Celeb-ID | | | | |
| Non-exemplar | 7.36E-3 | 8.44E-3 | 3.72 | 15.30 |
| Reference | 7.15E-3 | 7.97E-3 | 3.56 | 15.66 |
| Code | 7.00E-3 | 7.80E-3 | 3.77 | 14.62 |

\\begin{tabular}{| l | c c c c|}\n\\hline\nModel & L1 & MS-SSIM & Inception & FID \\\\\n\\hline\n\\multicolumn{5}{|c|}{Internal benchmark}\\\\\n\\hline\nNon-exemplar & 0.018 & 5.05E-2 & 3.96 & 11.27\\\\\nReference & 0.014 & 3.97E-2 & 3.82 & 7.67\\\\\nCode & 0.015 & 4.15E-2 & 3.94 ⋯⋯ \\end{tabular}

Figure A.8: Visualization examples of 7 tasks in DocGenome-test.

