# OpenReview forum: "DocGenome: A Large Benchmark for Multi-Modal Language Models in Real-World Academic Document Understanding"
_ICLR.cc/2025/Conference — Submitted to ICLR 2025_

### Official Review · Reviewer_i9VG · 2024-11-03

**Soundness:** 3
**Presentation:** 3
**Contribution:** 3
**Rating:** 8
**Confidence:** 4

**Summary:**

This work presents a large-scale multimodal academic document understanding dataset. It includes training and high-quality test sets, along with 7 benchmarking tasks proposed. The manuscript provides a clear description of the data collection and quality control processes. A series of large multi-modal models are benchmarked on the proposed test set, and training experiments are conducted to verify the effectiveness of the proposed training set.

**Strengths:**

1. The proposed large-scale multimodal academic document understanding dataset makes a solid contribution to the related research community, especially given the scarcity of such datasets.

2. The dataset collection and quality control are carefully designed, executed, and documented. The provided anonymous GitHub repository includes detailed documentation and links for downloading the dataset, supporting reproducibility and accessibility.

3. A training set is provided along with the test set, and training experiments are conducted to verify the effectiveness of the training set.

**Weaknesses:**

1. The QA data creation process depends heavily on GPT-4. As the QA pairs are generated by GPT-4, it can introduce biases regarding the type and difficulty of the questions. Further examinations into the potential biases would therefore be beneficial.

2. The evaluation metrics used for different tasks could be improved. Specifically, edit distance or BLEU may not accurately evaluate Equation-to-LaTeX and Table-to-LaTeX tasks, as these metrics do not account for the semantic equivalence of different LaTeX expressions. Additional evaluation could also be performed to verify the grammatical correctness of generated LaTeX expressions. Moreover, Open-ended QA tasks are evaluated using GPT-4 to compare reference and generated answers. While this is likely a reasonable approach, human evaluation to verify the reliability of GPT-4’s judgment would be beneficial.

**Questions:**

1. The manuscript discusses that the GPT-4-generated QA pairs are verified and updated by human annotators. What is the acceptance rate of the original QA pairs, or alternatively, what is the editing rate? Given that GPT-4 achieves around 60%-70% accuracy on QA tasks in Table 3, this suggests that a substantial portion of the QA pairs were likely updated.

2. Why are large multimodal models (e.g., GPT-4V, QWen-VL) not benchmarked on the Equation-to-LaTeX and Table-to-LaTeX tasks?

---

> ### Author Response · Authors · 2024-11-23
> **Reply to Reviewer i9VG**
>
> ***W1. Concerns about type and difficulty of the questions generated by GPT-4V***
>
> Thank you for your observation. As detailed in the appendix, we initialized QA pair generation with GPT-4 using a diverse set of QA examples for few-shot prompting. Additionally, Moreover, our quality control team, comprising 20 individuals with PhD or Master’s degrees, not only corrects answers but also edits incorrect questions to ensure the quality and difficulty of the QA pairs.
>
> ***W2. The evaluation metrics used for different tasks could be improved.***
>
> We also appreciate your suggestion, and we are indeed working on designing more reasonable evaluation metrics. We’ve provided [the figure](https://postimg.cc/dZ2Kymwx) to include our ongoing research on a novel metric design for the Equation-to-LaTeX task, which evaluates both LaTeX text similarity and region matching between the predicted LaTeX-rendered equation and the original image.
>
> ***Q1. More details about QA pairs generation and quality ensurance***
>
> In detail, the process is as follows:
> 1. GPT-4 was used to generate 7028 questions for 1757 paper samples.
> 2. Quality checkers first examine the questions, retaining or modifying them to obtain correct questions.
> 3. Each question was then allocated to two quality checkers for review and correction.
> 4. The checkers attempted to correct incorrect answers and assigned confidence scores.
> 5. Only QA pairs with the same answer and the highest confidence scores from both checkers were retained for the final dataset.
>
> Finally, 2498 QA pairs were retained to form the QA test set, of which 1672 were modified by the quality checkers. **The editing rate is 66.93%**
>
> We have already provided confidence score criteria in Appendix E.
>
> ***Q2. General VLMs on the Equation-to-LaTeX and Table-to-LaTeX tasks***
> We’ve included additional experiments with a general VLM on *Eqaution-to-LaTeX* and *Table-to-LaTeX* tasks. Taking Qwen2VL-7b as an example, its performance is not only inferior to the commercial closed-source model Mathpix but also falls short of our EqVLm and TableVLm trained on DocGenome.
>
> - **Eqaution-to-LaTeX:**
>
> |Model|Edit Distance|Jaccard Similarity|BLEU|Cosine Similarity|
> |-|:-:|:-:|:-:|:-:|
> |Mathpix|0.4738|0.7226|0.6045|0.4472|
> |Qwen2VL-7b|0.5824|0.6979|0.5506|0.1449|
> |EqVLM-B|**0.2111**|**0.8736**|**0.8621**|**0.6352**|
>
> - **Table-to-LaTeX:**
>
> |Model|Edit Distance|Jaccard Similarity|BLEU|Cosine Similarity|
> |-|:-:|:-:|:-:|:-:|
> |Mathpix|0.4436|0.7730|0.5826|0.3528|
> |Qwen2VL-7b|0.4876|0.7598|0.6979|0.4016|
> |TableVLM-B|**0.2223**|**0.8997**|**0.8800**|**0.5552**|
>
> At last but not least, we would like to take this opportunity to thank the anonymous reviewer again for the insightful comments and valuable suggestions, which greatly helped to improve the technical quality and the presentation of this manuscript. We sincerely hope that our response has addressed your concerns.

---

> > ### Comment · Reviewer_i9VG · 2024-11-23
> >
> > Thank you for your response, which addressed most of my concerns. It would be great if these responses could be reflected in the revised version (if they are not already). I've adjusted my assessment accordingly.

---

> > > ### Author Response · Authors · 2024-11-24
> > > **Response to Reviewer i9VG**
> > >
> > > We sincerely appreciate your thoughtful feedback and recognition of the contributions of our work. Thanks!

---

### Official Review · Reviewer_4Uws · 2024-11-03

**Soundness:** 3
**Presentation:** 3
**Contribution:** 2
**Rating:** 6
**Confidence:** 4

**Summary:**

The main contribution of this paper is a dataset called DocGenome that
is curated mostly automatically.

1.  The main component of paper is a dataset 500k samples of structured
    representation of academic papers. The author developed a pipeline
    to automatically parse the latex source.

2.  The authors also curate a test set of 7 tasks (vision-only,
    language-only, and multi-modal) to evaluate the performance of
    models on the dataset.

3.  The author also verified that the training data could be helpful and
    lead to better performance on downstream tasks via training on
    DocGenome data.

**Strengths:**

1.  The paper is nicely written, and there authors have conducted a lot
    of experiments evaluating different aspects of this dataset.

2.  The dataset, as well as the document conversion pipeline, could be a
    helpful resource for the community.

**Weaknesses:**

1.  Overall this paper feels like a resubmission from venues like the
    NeurIPS dataset and benchmark track. The methodology contribution is
    relatively weak: the method of creating the dataset (i.e., parsing
    latex sources) is not novel (e.g., a lot of earlier work like GROTOAP, PubLayNet, DocBank, all use similar approaches ),
    neither is the framing of the tasks (all these tasks have been
    studied before to some extent). I am inclined to a borderline reject
    unless the authors can give a strong statement in terms of the
    novelty of the dataset or the methodology.

2.  While I find there could be one potentially novel aspect of the
    dataset---the "13 categories of component units and 6 types of
    logical relationships between them", it is not clear how the authors
    actively use this component in the experiments. (The empirical
    studies mostly focus on using individual components but not the
    relationships.) I'd encourage the authors to provide more
    insights/experiments on this.

3.  There's another limitation of the datasets: since it is curated via
    automatically parsing latex sources, the variance and the structure
    could be limited, and the trained models might not be able to
    transfer to other types of papers (which is also mentioned by the
    authors in line 458.)

**Questions:**

1.  The author mentioned that "Each QA pair is reviewed by three
    reviewers for cross-verification." (line 310) It would be great if
    the authors can provide more details in terms of the inter-annotator
    agreement or additional details. "Finally, the two
    manually-evaluated results, along with the automatically-evaluated
    result are cross-verified with the original text to ensure accuracy
    and consistency" (line 317) I am not sure how this process works,
    can you provide more details?

---

> ### Author Response · Authors · 2024-11-23
> **Reply to Reviewer 4Uws: Part-2**
>
> ***W2. More insights/experiments on layout information and logical relationships***
>
> We deeply appreciate the reviewer’s thoughtful feedback and the time and effort dedicated to evaluating our paper. We propose the following insights and experiments:
>
> - **Better layout detection aids in document understanding:** We have already included this part in Section 5.4 of our original paper. For questions related to figures or tables, we directly annotated the detection boxes for figures or tables on the document images to serve as the image input for the QA task. Taking internVL-1.5 as an example, when the images of the papers contain layout information (this information can be considered as a prompt) relevant to the questions, the performance of the QA task can be further enhanced.
>
> |Model|#Params|With Layout Information in Imgae|Sigle-Page QA|Multi-Page QA|
> |-|:-:|:-:|:-:|:-:|
> |InternVL-1.5|26B|&#10007;|0.4529|0.3577|
> |InternVL-1.5|26B|&#10004;|**0.4922**|**0.4030**|
>
> - **The relationships between document elements facilitate the expansion of multimodal data:**
> Utilizing the relation information between entities, we can index associated modal information for specific tasks, enabling data re-annotation and expansion. For instance, in constructing table-related QA tasks, we can not only obtain images of tables but also index the text describing the tables in the document, thereby enriching multimodal information for re-annotation and task expansion. The relationships make our annotation information more flexible and actionable.
>
>
> ***W3.  Concerns about variance and the structure of document due to LaTeX***
>
> We sincerely appreciate the reviewer’s comments and the valuable time spent reviewing our paper. As mentioned by the reviewer, in Table 6 (Line 458), Mathpix demonstrates slightly better performance on Out-Of-Distribution (OOD) data, compared to the model trained using our DocGenome, e.g., 0.4873 vs. 0.6627.
>
> However, a direct comparison between our model and Mathpix may not be entirely fair, as Mathpix is a commercial software developed with an investment of hundreds of millions of dollars. **In contrast, our DocGenome is an automated annotation pipeline that constructs datasets without incurring any cost**. On the other hand, the diversity of our proposed DocGenome dataset in scientific literature is well ensured, thanks to the wide range of document layouts provided by the arXiv open-access community, which covers 153 academic disciplines. To validate this diversity, we conducted experiments from two key perspectives, as outlined below:
>
> - **Page-level diversity**: Supplementary materials Figure A.1 demonstrate the diversity of the page-level data distribution.
> - **Layout diversity**: Table 6 presents experiments conducted on a different domain, SciHub. The results demonstrate that models trained using the layout information from DocGenome outperform DocXChain and achieve significant improvements. This further highlights the diversity of the layout information in the DocGenome dataset.
>
>
> ***Q1. More details in terms of the inter-annotator agreement or additional details.***
>
> In detail, the process is as follows:
> 1. GPT-4 was used to generate 7028 questions for 1757 paper samples.
> 2. Quality checkers first examine the questions, retaining or modifying them to obtain correct questions.
> 3. Each question was then allocated to two quality checkers for review and correction.
> 4. The checkers attempted to correct incorrect answers and assigned confidence scores.
> 5. Only QA pairs with the same answer and the highest confidence scores from both checkers were retained for the final dataset.
>
> Finally, 2498 QA pairs were retained to form the QA test set, of which 1672 were modified by the quality checkers.
>
> We have already provided confidence score criteria in Appendix E.
>
>
> Thank you once again for your valuable feedback. We have revised our paper based on your suggestions, with all modifications highlighted in orange. Your input has been immensely helpful in enhancing the quality of our work.

---

> > ### Author Response · Authors · 2024-11-24
> > **Reply to Reviewer 4Uws: Part-1**
> >
> > | Modality Class | Amount of Data | Data Form | Data Description|
> > |-|:-:|:-:|:-:|
> > |Algorithm|112,706| Image-LaTex pairs | algorithmx package in LaTex |
> > |Image-Caption | 4,404,530 | Image-Text pairs | Images in paper and the corresponding caption|
> > |Figure-TiKZ |4,612,156| Image-TiKZ code pairs |Flowcharts, geometric diagrams, and other images compiled using tikzpicture along with their corresponding TiKZ representations |
> > | Equation-Latex | 11,784,597 | Image-LaTex pairs | Displayed formula |
> > | Footnote |22,957| Image-Text pairs |Footnotes appearing in the paper |
> > | List | 905,843 | Image-LaTex pairs | Listed in an enumerated format |
> > | Table | 678,931 | Image-LaTex pairs | Tables formatted in LaTeX |
> > | Text | 16,310,161 | Image-Text pairs | Plain text regions in the paper |
> > | Title | 6,191,633 | Image-LaTex pairs | Headings at all levels |
> > | Text-EQ | 27,282,610 | Image-LaTex pairs | Text regions that includes inline formulas |
> > | Code | 50,849 | Image-Code pairs | Actual code regions, such as Python, etc |
> > | Abstract | 689,449 | Image-Text pairs | Abstract region in the paper |
> >
> > ----------------
> >
> >
> > We sincerely appreciate your review and feedback on this paper. Although similar works (such as GROTOAP, PubLayNet, and DocBank) have also adopted methods to parse LaTeX source code, ***we would like to further clarify the uniqueness and contributions of DocGenome through the following points***:
> >
> > - As summarized in the table above, our proposed DocGenome not only captures the layout information of each modality but also allows for the customized extraction of each complex modality, such as Table, Equation, Text-EQ, Title, and Abstract. This approach comprehensively preserves all the information in a paper and facilitates the development of downstream applications, such as [equation extraction](https://github.com/opendatalab/UniMERNet), [document content extraction](https://github.com/opendatalab/MinerU), [table content extraction](https://github.com/UniModal4Reasoning/StructEqTable-Deploy) [automatic survey](https://github.com/AutoSurveys/AutoSurvey), etc.
> >
> > - By utilizing Table-latex pairs and Equation-latex pairs, we can develop effective parsers that convert table and equation images into markdown or LaTeX source code. For example, **to the best of our knowledge, the recently-popular open-source repository [minerU](https://github.com/opendatalab/MinerU) leverages our DocGenome dataset to achieve this functionality.**
> >
> > - Using the Title and Abstract, we can conduct research on [automatic paper survey](https://github.com/AutoSurveys/AutoSurvey) task,  aiming to generate high-quality academic surveys based on the title and abstract of a given paper.
> >
> > - DocGenome also includes multiple modality categories, such as flowcharts and Image-caption pairs (shown in [this figure](https://postimg.cc/Vd92rS2n)), which can serve as corpora for tools like image-to-text generation tasks and geometric question-answering models.
> >
> > - Compared to prior works like GROTOAP, PubLayNet, and DocBank, our proposed DocGenome offers a significantly larger dataset (**6.8 million page-level structured data points** and **72 million region-level structured data points**) along with more diverse structured representations, such as Table-latex and Figure-TiKZ. To the best of our knowledge, it stands as the largest and most diverse scientific document dataset in terms of both scale and variety.
> >
> > Therefore, DocGenome demonstrates significant advantages in terms of data quantity, data diversity, data value, and data quality. We believe that DocGenome will serve as a critical resource for advancing multimodal document parsing and related research areas, providing strong support to the community and inspiring more innovative applications and research outcomes in the future.
> >
> > ***To address your concerns, we have further revised our paper and marked the changes in orange text.***
> >
> > At last but not least, we would like to take this opportunity to thank the anonymous reviewer again for the insightful comments and valuable suggestions, which greatly helped to improve the technical quality and the presentation of this manuscript. We sincerely hope that our response has addressed your concerns. We would greatly appreciate it if you could reconsider the contributions of our proposed DocGenome to the community and its potential value for various academic research tasks in the future.

---

### Official Review · Reviewer_TmKL · 2024-11-04

**Soundness:** 3
**Presentation:** 3
**Contribution:** 3
**Rating:** 5
**Confidence:** 5

**Summary:**

Long  document understanding is an interesting and complex problem. The paper provides a document processing pipeline and a large and diverse collection of documents for academic document understanding. The paper also evaluates the performance of large models on multiple tasks using the benchmark dataset.

**Strengths:**

As shown in Table 1 of the paper, the document collection is superior to existing collections on multiple aspects.

**Weaknesses:**

1. While datasets and benchmarking are important to drive research, the paper does not contain research contributions.
2. The documents are based on latex formatted documents which is a subclass of all documents.

**Questions:**

1. Is the collection representative of scientific documents? In other words, does latex formatting constrain documents in some way? For example, latex denotes non-existing references by ??. If a document is created using other formatting software, such mistakes are not flagged.

---

> ### Author Response · Authors · 2024-11-23
> **Reply to Reviewer TmKL**
>
> We deeply appreciate the reviewer’s thoughtful feedback and the time and effort dedicated to evaluating our paper. We will address the reviewer’s concerns regarding the effectiveness of using LaTeX format in the following part.
>
> LaTeX is the de facto standard for scientific writing, particularly in STEM fields, due to its ability to produce professional, high-quality documents. It is widely adopted by top journals and conferences, such as IEEE, ACM, and Springer. **Note that** many top journals and conferences require or recommend submitting manuscripts in LaTeX.
>
> LaTex format has the following key advantages:
>
> - **Superior Mathematical Typesetting**: LaTeX excels in handling complex equations, ensuring precision and aesthetic consistency.
>
> - **Extensibility**: Thousands of packages (e.g., amsmath, graphicx, biblatex) allow customization for various academic needs.
>
> - **Graphics and Visualization**: Tools like TikZ and PGFPlots enable seamless creation of high-quality, scalable graphics and plots. By leveraging the proposed DocGenome, we can extract the TiKZ code and the corresponding rendered image, as illustrated [in this figure](https://postimg.cc/Vd92rS2n).
>
> - **Portability and Collaboration**: As a plain-text system, LaTeX supports version control (e.g., Git) and works across platforms (Windows, macOS, Linux).
>
> **On the other hand**, the arXiv community hosts papers under **the CC license**, and all papers are represented in LaTeX format. As a result, the structured scientific literature we obtain based on arXiv (using LaTeX code) also complies with the CC license, which helps to widely promote the dissemination and use of our open-source dataset (DocGenome). Moreover, by leveraging LaTeX code, we can automatically extract annotated structures from 600,000 scientific papers without incurring any human-annotation costs.
>
> **Overall, the collection is representative of scientific documents**, as LaTeX is the preferred tool for academic writing in STEM fields due to its precision and professional formatting. While LaTeX does impose strict formatting rules, these are not constraints but rather mechanisms to ensure accuracy and consistency. **For instance, the "??" marker for missing references serves as a clear indicator of errors, prompting authors to address them before finalizing the document**. This feature actually enhances the quality of scientific writing by reducing the likelihood of overlooked mistakes. In contrast, other formatting software may not flag such issues, potentially leading to incomplete or inconsistent references. As a result, LaTeX’s strictness contributes to its reputation as a standard for rigorous scientific documentation.
>
>
> At last but not least, we would like to take this opportunity to thank the anonymous reviewer again for the insightful comments and valuable suggestions, which greatly helped to improve the technical quality and the presentation of this manuscript. We have further supplemented some descriptive languages regarding LaTeX's representational ability in the main text. Hope our response has addressed your concerns.
>
>
> It is appreciated that you could reconsider the representational ability and extensibility of our work, as well as our contributions to this community. Thank you.

---

> > ### Comment · Reviewer_TmKL · 2024-11-25
> > **Thanks, and ...**
> >
> > I thank the authors for responding to my feedback. The authors have highlighted the central role LaTeX plays in scientific publishing. My question was not on the importance of LaTeX but on how representative LaTeX created articles are. Per arXiv's statement "arXiv is a free distribution service and an open-access archive for nearly 2.4 million scholarly articles in the fields of physics, mathematics, computer science, quantitative biology, quantitative finance, statistics, electrical engineering and systems science, and economics." This covers 8 areas. There are many more areas which are not covered, civil engineering, chemistry and materials science to name a few.
> >
> > Even in these areas covered by arXiv, the conferences and journals provide both LaTeX and Word templates.
> >
> > I believe there is a misunderstanding in understanding my question "Is the collection representative of scientific documents?".
> >
> > In full disclosure, I have used LaTeX for all my papers and my comment does not stem from ignorance or under-appreciation of LaTeX.
> >
> > I once again thank the authors for responding to my comment.

---

> ### Author Response · Authors · 2024-11-25
> **Reply to Reviewer TmKL, Round 2**
>
> Dear Reviewer TmKL,
>
> Thank you for your continued engagement and for clarifying your concerns regarding our manuscript. We understand that your question centers on the representativeness of LaTeX-created articles across all scientific disciplines, rather than the importance of LaTeX itself.
>
> We acknowledge that while arXiv covers a substantial number of fields, it does not encompass all areas of research, such as civil engineering, chemistry, and materials science. We appreciate your point that conferences and journals in these fields often offer both LaTeX and Word templates, which reflects the diversity of preferences within the scientific community.
>
> To address your concern about representativeness, we would like to emphasize that our work primarily focuses on fields where LaTeX is predominantly used, as evidenced by its widespread adoption in journals and conferences within these domains. While our dataset may not cover every scientific discipline, it provides a comprehensive representation of those areas where LaTeX is the standard.
>
> Regarding your question, **"how representative LaTeX-created articles are,"** we have provided a specific implementation plan in the main text as follows:
>
> - For disciplines not covered by the arXiv open-access community, such as civil engineering, chemistry, and materials science, we assume that we can collect their PDF data;
>
> - We can leverage the models **trained using our proposed DocGenome dataset** to perform the inference processing on these out-of-distribution disciplines, such as civil engineering, chemistry, and materials science, thereby achieving generalization to these datasets;
>
> - Actually, **we have conducted additional experiments to evaluate the generalization ability of our approach to other disciplines**. Specifically, we collected PDF data from the Scihub domain, which encompasses a broader range of scientific fields, including civil engineering, chemistry, and materials science not fully represented in arXiv. Using our DocGenome-trained models, we applied our scientific document parsing method to the Scihub domain. As shown in Table 6 of the main text, our method has demonstrated strong performance in scientific document parsing for Scihub domain, even when applied to diverse disciplines outside the core focus of arXiv. Notably, our approach outperforms existing layout detection tool, such as DocXChain, in terms of accuracy and robustness.
>
>
> | Model | mAP@0.5:0.95↑ | Title | Text | Figure  | Caption  | Equation | Table  | Footnote |
> |-|:-:|:-:|:-:|:-:|:-:|:-:|:-:|:-:|
> |DocXChain (Open-source Toolchain for Document parsing) | 37.99 | 32.53 | 59.00 | 67.17 | 38.71 | 12.98 | 38.99 | 16.54 |
> |YOLOv8 (Trained using the proposed DocGenome) | 50.15 | 42.59 | 64.87 | 56.65 | 64.51 | 47.14 | 47.08 | 28.21|
>
> Table Caption: Results on Scihub domain, which shows the generalization ability of the proposed DocGenome on other disciplines outside the core focus of arXiv.
>
> Finally, **we would like to emphasize** that we also recognize the value of diverse documentation tools and formats in scientific publishing. Our intention is not to diminish the importance of other formats but to highlight the strengths and capabilities of LaTeX in the contexts where it is most prevalent.
>
> We have carefully revised our paper in response to your second-round questions and highlighted the changes in orange. Please refer to page 9 and Table 6. Looking forward to your response and further discussion.

---

> > ### Comment · Reviewer_TmKL · 2024-11-25
> > **SciHub and DocXChain**
> >
> > Thanks for providing the clarifying information. I have the following questions.
> >
> > 1. What are the details of SciHub dataset? Do we know which scientific fields are represented in the dataset? Also, any sense for overlap with arXiv?
> >
> > 2. DocXChain: DocXChain is a useful tool but not a peer-reviewed research paper, as far as I can tell. How do we know if it represents the state-of-the-art?

---

> ### Author Response · Authors · 2024-11-25
> **Reply to Reviewer TmKL, Round 3**
>
> Dear Reviewer TmKL,
>
> Thank you for your further insightful questions. We appreciate the opportunity to clarify the details regarding the SciHub dataset and our comparison with DocXChain.
>
> >***Q1. SciHub Dataset Details***
>
> We sampled the entire SciHub dataset and created a subset of 966 papers, covering disciplines such as medicine, chemistry, biology, and humanities **as shown in the following table**, in order to evaluate the generalization capability of our models across a wider spectrum of disciplines. While the exact distribution of fields within the SciHub dataset varies, it provides a diverse representation that complements the areas covered by arXiv. [This figure](https://postimg.cc/G8NQ6sLX) visualizes some paper examples from our constructed SciHub dataset.
>
> Regarding overlap with arXiv, only a very small portion of disciplines, such as computer science, intersect with arXiv's fields, and these account for **less than 3% of the data**. On the other hand, as you mentioned, our primary focus was on testing our approach in scientific fields where arXiv's coverage is highly limited, thereby highlighting the versatility and robustness of our document parsing method.
>
> ||Medicine|Chemistry|Biology|Humanities|Physics|Engineering|Math|Ecology|Computer Science|Economics|Geography|
> |-|:-:|:-:|:-:|:-:|:-:|:-:|:-:|:-:|:-:|:-:|:-:|
> |Amount|237|159|150|121|84|67|36|35|27|25|25|
> |Proportion|24.53%|16.46%|15.53%|12.53%|8.70%|6.94%|3.73%|3.62%|2.80%|2.59%|2.59%|
>
> Table: The discipline distribution of the 966 papers from our constructed Scihub dataset
>
>
> >***Q2. Comparison with DocXChain***
>
> DocXChain, a highly popular tool currently, is designed to handle a wide range of open-domain document parsing tasks. While it is a practical tool rather than a peer-reviewed research publication, our goal was to benchmark its performance against existing tools, demonstrating significant improvements in accuracy, particularly for papers outside the arXiv domain. This comparison is specifically designed to address the question of whether our approach can achieve a certain level of effectiveness, **when applied to scientific documents from other disciplines**. It is indeed evident from Table 6 of the main text, that our method can be applied to the Scihub domain.
>
> We have supplemented the corresponding dataset information in the [revised paper](https://openreview.net/pdf?id=CI9JMBAsPg), and please refer to Table 7 in our main text. We hope this additional information addresses your questions and provides a clearer understanding of our work. We are grateful for your feedback, which has been invaluable in refining our manuscript.
>
> Thank you once again for your thoughtful review and consideration.
>
> &nbsp;
>
> Best regards,
>
> Authors

---

### Official Review · Reviewer_c3jL · 2024-11-06

**Soundness:** 3
**Presentation:** 3
**Contribution:** 3
**Rating:** 6
**Confidence:** 4

**Summary:**

The paper proposed a new dataset for academic document understanding. By processing 500k documents from arXiv (using their source LaTeX files) the authors create a new large dataset which covers diverse disciplines, preserves data in all modalities from papers, and covers 7 document-oriented tasks.
The paper describes its automatic processing and labeling pipeline, along with the two metrics used for quality assurance. The authors then describe how the dataset is split into train and test subsets; where the data is divided into tiers based on the previously mentioned metrics and 1004 papers are sampled from the top tier.
The sampled papers are them used to create QA pairs about their content (both single-page and multi-page questions) using GPT-4V, which are them validated by professional faculty members.
The paper also present the benchmarking of different LLMs on its test set, along with the usage of its training data to effectively create new models that outperform selected baselines.

**Strengths:**

The paper presents a new large dataset of multi-modality academic documents for document-understanding tasks. This is a welcome and useful target as most existing datasets are limited in scope, diversity, and especially do not preserve all modalities of data in the source papers, losing important information.

The process to create the dataset is well presented and pragmatic and both the toolset and processed data can be valuable in themselves or for further collections and annotations for new tasks.

While somewhat small when compared to the overall collected data, the annotated test set with questions is shown to be already a useful benchmark for recent large multi-modality models.

**Weaknesses:**

Unfortunately, the dataset also presents somewhat limited novelty and impact.

In its current described form, one of its main contributions is the created test set with annotated QAs. However, these seem to have limited coverage of non text modality QA and the annotated relationships are limited to layout level, not focusing on critical data/information relationships. As a dataset targeting multi-modality models, it is critical that these are emphasized.

This is especially the for charts/plots and tables, which are rich in information and relationships. I missed seeing some analysis on the modalities and specific discussions on issues and how the dataset addresses them.

**Questions:**

With so many different disciplines covered in the data, how exactly were the faculty members selected to review the annotated dataset?

Will the data and codebase be released under a permissive license? With the lack of explicit table-QA or image-QA/chart-QA annotations, for example, it would be critical that the data can be used for re-annotation and extensions.

In line 161 you say \ref commands are removed, but these seem essential in relationship extraction. Also, Table 2 implies they are indeed used.

If GPT-4V was used to create all questions, is there a reason its performance in Table 3 is not so high and even worse than GPT-4o?

How many QA pairs were actually created and used? My understanding is that 4 QA pairs are created per sample, but only 3k total QA pairs were kept? Did I miss some details on how this was filtered?

Please provide more details on the sampled data for the OOD experiments. This data should also be released for reproducibility.

---

### Author Response · Authors · 2024-12-03
**ICLR Reminder of Reviewer 4Uws**

Dear Reviewer 4Uws,

We have thoroughly addressed your concerns and look forward to your feedback, which will greatly assist us in making any necessary revisions or clarifications.

Thank you for your time and consideration.

Best regards,
Authors of paper 6842

---

### Note · Authors · 2024-11-28

I have read and agree with the venue's withdrawal policy on behalf of myself and my co-authors.

---

> ### Note · Program_Chairs · 2024-11-28
>
> We approve the reversion of withdrawn submission.

---

### Meta-Review · Area_Chair_diNG · 2024-12-23

**Metareview:**

This paper introduces DocGenome, a dataset curated through automated methods consisting of a large number of samples representing the structured content of academic papers. To create this dataset, it describes a pipeline developed to automatically parse LaTeX source files. To evaluate the dataset, the paper includes a test suite of seven tasks. These tasks cover vision-only, language-only, and multi-modal challenges, providing a framework to assess model performance on DocGenome.The paper also demonstrates that training on DocGenome data can improve model performance on downstream tasks, showing its utility for research in document understanding and related fields.

Reviewers generally found the paper well-written and the released dataset valuable for the research community. They appreciated the quality control measures implemented during dataset creation and acknowledged the usefulness of the training set.

However, a significant critique focused on the paper's methodological and technical contributions, as well as the dataset's novelty, which reviewers considered relatively limited (4Uws, c3jL, TmKL). Some reviewers also expressed concerns regarding limited coverage of non text modalities (c3jL).  Concerns were also raised about the automated construction process of the dataset, particularly its reliance on GPT-4 (i9VG, 4Uws). Additional issues included a lack of experiments addressing novel aspects of the dataset, potential biases in its creation, the emphasis on LaTeX formatting, and requests for further details—all of which were largely resolved during discussions.

Overall, while the resource offers value to the community, its technical contributions and novelty appear limited. The authors' response summarized the weaknesses raised by reviewers as a separate comment, but notably omitted addressing these major critiques.

**Additional Comments On Reviewer Discussion:**

Reviewers raised questions about the bias and diversity of the data created by LLMs. In response the authors performed additional experiments.
Reviewers asked for additional details on the specifics of data quality measures, the annotators, parts of the pipeline (e.g., QA generation), and the OOD dataset. The author response addresses all these issues.
In response to one question about details of faculty used for data inspection, the response provides exact number and hours, but does not provide details of the approximate range of compensation used for data collection.
In response to questions about additional experiments, the response includes several additional experiments (e.g., out of domain generalization, and experiments on Eqaution-to-LaTeX and Table-to-LaTeX tasks).

---

### Decision · Program_Chairs · 2025-01-22

Reject